# REEVALUATING POLICY GRADIENT METHODS FOR IMPERFECT-INFORMATION GAMES

**Max Rudolph**[*,1]**, Nathan Lichtlé**[*,2]**, Sobhan Mohammadpour**[*,3]**, Alexandre Bayen**[2]**,
J. Zico Kolter**[4]**, Amy Zhang**[1]**, Gabriele Farina**[3]**, Eugene Vinitsky**[5]**, Samuel Sokota**[4]
[1]University of Texas at Austin, [2]University of California, Berkeley,
[3]Massachusetts Institute of Technology, [4]Carnegie Mellon University,
[5]NYU Tandon School of Engineering
`mrudolph@cs.utexas.edu`, `nathan.lichtle@gmail.com`, `somo@mit.edu`,
`gfarina@mit.edu`, `vinitsky.eugene@gmail.com`, `ssokota@andrew.cmu.edu`
[*]Equal contribution

## ABSTRACT

In the past decade, motivated by the putative failure of naive self-play deep reinforcement learning (DRL) in adversarial imperfect-information games, researchers have developed numerous DRL algorithms based on fictitious play (FP), double oracle (DO), and counterfactual regret minimization (CFR). In light of recent results of the magnetic mirror descent algorithm, we hypothesize that simpler generic policy gradient methods like PPO are competitive with or superior to these FP-, DO-, and CFR-based DRL approaches. To facilitate the resolution of this hypothesis, we implement and release the first broadly accessible exact exploitability computations for five large games. Using these games, we conduct the largest-ever exploitability comparison of DRL algorithms for imperfect-information games. Over 7000 training runs, we find that FP-, DO-, and CFR-based approaches fail to outperform generic policy gradient methods.

## 1 INTRODUCTION

An imperfect-information game (IIG) is one in which there is information asymmetry between players or in which two players act simultaneously. Two-player zero-sum IIGs—those in which two players with opposite incentives compete against one another—are among the most well-studied IIGs. This is in part due to the fact that, for these games, there is a sensible and tractable objective: minimizing the amount by which a player can be exploited by a worst-case adversary.

A natural approach to model-free deep reinforcement learning (DRL) in such games is to deploy a single-agent algorithm in self-play. However, because imperfect information induces cyclical best response dynamics, such an approach can fail catastrophically, yielding policies that are maximally exploitable. So as to avoid this outcome, most existing literature focuses on adapting pre-existing tabular algorithms with established equilibrium convergence guarantees, such as fictitious play (FP) (Brown, 1951; Robinson, 1951), double oracle (DO) (McMahan et al., 2003), and counterfactual regret minimization (CFR) (Zinkevich et al., 2007), to DRL.

Unfortunately, constructing effective DRL approaches from these tabular algorithms has proven challenging. FP and DO require expensive best response computations at each iteration (i.e., each iteration requires solving an entire reinforcement learning problem) and can exhibit slow convergence as a function of iteration number (Zhang and Sandholm, 2024). While CFR converges more quickly, its immediate adaptation to model-free reinforcement learning requires importance weighting over trajectories (Lanctot et al., 2009), resulting in high variance feedback to which function approximation is poorly suited. Moreover, none of FP, DO and CFR enjoy last-iterate convergence, creating a layer of indirection to extract the desired policy.

Recently, Sokota et al. (2023) demonstrated the promise of an alternative algorithm, a policy gradient (PG) method called magnetic mirror descent (MMD). They showed that MMD achieves competitive

---

Code available at `https://github.com/nathanlct/IIG-RL-Benchmark` and `https://github.com/gabrfarina/exp-a-spiel`.

performance with CFR in tabular settings, while retaining the deep learning compatibility native to reinforcement learning algorithms.

In this work, we expound the relationship between MMD and other generic deep PG methods such as PPO (Schulman et al., 2017). Like MMD, the improvement steps of these other generic PG methods 1) maximize expected value, 2) regularize the policy, and 3) control the size of the update. These parallels lead us to the question: Given that MMD performs well as a DRL algorithm for IIGs, shouldn't these other generic PG methods perform well, too?

We believe that the answer to this question is yes. To further this idea, we put forth the following hypothesis:

> *With proper tuning, generic PG methods are competitive with or*
> *superior to FP-, DO-, and CFR-based DRL approaches in IIGs.*

The confirmation of this hypothesis would have large ramifications both for researchers and practitioners. It would force researchers who currently dismiss approaches like PPO to revisit fundamental beliefs. And, for practitioners, it would justify the use of simpler PG methods, potentially offering both cleaner implementations and improved performance.

However, assessing the above hypothesis is not straightforward. Doing so demands exact exploitability computations for games large enough to be representative of settings to which DRL is relevant, but small enough that such computations are efficient. Unfortunately, due to the complexity of implementing these computations for large games, few are publicly available, even in OpenSpiel (Lanctot et al., 2019)—the standard library for DRL in IIGs. This issue has plagued the field, forcing researchers to resort to tenuous metrics and making progress difficult to measure.

We address this issue by implementing exact exploitability computation for five large games: 2D5F Liar's Dice, 3x3 Dark Hex, Phantom Tic-Tac-Toe, 3x3 Abrupt Dark Hex, and Abrupt Phantom Tic-Tac-Toe. These games have millions of infostates—tens of thousands more than many games for which exploitability is typically reported, such as Leduc Hold'em (Southey et al., 2005).

Using these benchmarks, we undertake the largest-ever exploitability comparison of DRL algorithms for IIGs, spanning 7000 total runs across 350 hyperparameter configurations, requiring over 345,000 CPU hours, and including NFSP (Heinrich and Silver, 2016), PSRO (Lanctot et al., 2017), ESCHER (McAleer et al., 2023), R-NaD (Perolat et al., 2022), MMD (Sokota et al., 2023), PPO (Schulman et al., 2017) and PPG (Cobbe et al., 2021). Over these runs, all of which are included in the paper, NFSP, PSRO, ESCHER, and R-NaD fail to outperform the generic PG methods (MMD, PPO, PPG).

We release our implementations for computing exact exploitability and head-to-head values, which require under 2 minutes on commodity hardware, with OpenSpiel-compatible Python bindings, as well as efficient implementations of tabular solvers and best responses. We encourage the community to use these tools both to test our hypothesis and as benchmarks for future research.

## 2 PRELIMINARIES

To start, we set context by giving a formalization of IIGs and exploitability, and an overview of model-free DRL approaches to IIGs.

### 2.1 GAME FORMALISM

IIGs are often formalized as perfect-recall extensive-form games (EFGs) (Kuhn, 2003). Our formalism is equivalent to perfect-recall EFGs, but possesses greater superficial similarity to traditional reinforcement learning notation. We notate a game as a tuple $\langle \mathbb{S}, \mathbb{A}, \mathbb{O}, \mathbb{I}, \mathcal{R}, \mathcal{T}, \mathcal{O}, \mathcal{C}, t_{\max} \rangle$, where $\mathbb{S}$ is the set of game states; $\mathbb{A}$ is the set of actions; $\mathbb{O}$ is the set of observations; $\mathbb{I} = \cup_t (\mathbb{O} \times \mathbb{A})^t \times \mathbb{O}$ is the set of information states; $\mathcal{R} \colon \mathbb{S} \times \mathbb{A} \to \mathbb{R}$ is the reward function for player 1 (and the cost function for player 2); $\mathcal{T} \colon \mathbb{S} \times \mathbb{A} \to \Delta(\mathbb{S})$ is the transition function; $\mathcal{O} \colon \mathbb{S} \to \mathbb{O}$ is the observation function, which determines the acting player's observation; $\mathcal{C} \colon \mathbb{S} \to \{1, 2\}$ is the player choice function, which determines the acting player; $t_{\max}$ is the time step at which the game terminates.

The players interact with the game using policies $\pi_i \colon \mathbb{I} \to \Delta(\mathbb{A})$ for $i \in \{1, 2\}$ mapping information states to distributions over actions.

The objective of player 1 and loss of player 2 is the expected return $\mathcal{J}(\pi) = \mathbb{E}_\pi \sum_{t=1}^{t_{\max}} \mathcal{R}(S^t, A^t)$.

Directly optimizing the expected return is only possible given a particular opponent or distribution of opponents. In practice, identifying this distribution may be prohibitively expensive or altogether infeasible—such a distribution may not even exist.

The standard resolution to this issue is the pursuit of minimax guarantees. The metric associated to minimax guarantees is called exploitability:

$$\mathrm{expl}(\pi) = \frac{\max_{\pi_1'} \mathcal{J}(\pi_1', \pi_2) - \min_{\pi_2'} \mathcal{J}(\pi_1, \pi_2')}{2}.$$

Exploitability is the expected return of a worst-case opponent, called a best response, playing half of the time as player 1 and half as player 2. Joint policies with exploitability zero are Nash equilibria.

Exploitability is an attractive resolution both because: 1) if exploitability zero is achieved, no opponent can do better than tie in expectation (assuming the player plays each role half of the time); and 2) in complex games, it tends to be the case that even achieving low, positive exploitability leads to reliably winning against opponents in practice.

## 2.2 Algorithmic Approaches

Existing model-free DRL approaches to producing low exploitability policies for two-player zero-sum IIGs are largely based on one of naive self-play, best responses, CFR, and regularization.

### 2.2.1 Naive self-play

Of these approaches, naive self-play is appealing for its simplicity; however, it is easy to show that it can produce catastrophic results. Some value-based algorithms cannot even express non-deterministic policies. This makes it impossible for them to achieve low-exploitability in many games (e.g., in rock-paper-scissors, deterministic policies are maximally exploitable). While PG methods can express non-deterministic policies, their learning dynamics generally cycle, diverge or exhibit chaotic behavior, rather than converge to Nash equilibria (Cheung and Piliouras, 2019).

### 2.2.2 Best Responses

A next-simplest alternative is to leverage tabular game-theoretic algorithms that rely on best-response computations. These game-theoretic algorithms support a plug and play workflow with model-free DRL, by substituting the exact best response for an approximate best response computed via DRL. And, unlike naive self-play, they come with convergence guarantees.

Fictitious play (FP) (Brown, 1951; Robinson, 1951) is one such game-theoretic algorithm. At each iteration $t$, FP computes a best response $\pi_t$ to the average of the previous iterations $\bar{\pi}_{t-1} = \mathrm{avg}(\pi_1, \ldots, \pi_{t-1})$. This average is over the sequence form of the policy, meaning that it is equivalent to a policy that uniformly samples $T \sim \mathrm{uniform}(\{1, \ldots, t-1\})$ at the start of each game and then plays according to $\pi_T$ until the conclusion of the game. The exploitability of the average $\bar{\pi}_t$ of a sequence of policies generated by FP converges to zero as $t$ grows large.

Double oracle (DO) (McMahan et al., 2003) is another such game-theoretic algorithm. Like FP, DO works by iteratively computing best responses. However, rather than computing best responses to the average of the previous iterates, DO computes best responses $\pi_t$ to Nash equilibria $\sigma_t^*$ of a so-called metagame. This metagame is a one-shot game in which players select policies $\pi^{(1)}, \pi^{(2)}$ from among the previous best responses $\{\pi_1, \ldots, \pi_{t-1}\}$ and receive as payoff the expected value $\mathcal{J}(\pi^{(1)}, \pi^{(2)})$ of playing these policies against one another. A policy of this metagame is a mixture $\sigma_t$ over previous best responses $\{\pi_1, \ldots, \pi_{t-1}\}$ that is enacted by sampling $\pi_T \sim \sigma_t$ once at the start of each game, then playing according to $\pi_T$ until the conclusion of the game. A policy of this metagame $\sigma_t$ is unexploitable if none of the previous best responses $\{\pi_1, \ldots, \pi_{t-1}\}$ exploits it. The exploitability of a sequence of metagame Nash equilibria is guaranteed to converge to zero as $t$ grows large.

Unfortunately, both FP and DO can suffer from slow convergence. Requiring a large number of iterations is especially concerning in the context of DRL because each iteration requires solving

an entire reinforcement learning problem. Despite numerous works on DRL variants of both FP (Heinrich and Silver, 2016; Qin et al., 2019; Vinyals et al., 2019; Goldstein and Brown, 2023; Hu et al., 2023; Li et al., 2023a) and DO (Lanctot et al., 2017; McAleer et al., 2020; 2021; Smith et al., 2021; McAleer et al., 2022a; Dinh et al., 2022; Zhou et al., 2022; Yao et al., 2023; Tang et al., 2023; McAleer et al., 2024; Li et al., 2024a; Lian, 2024), it is not clear that this issue has been addressed.

### 2.2.3 Counterfactual Regret Minimization

Counterfactual regret minimization (CFR) (Zinkevich et al., 2007), widely regarded as the gold standard for tabularly solving IIGs, generally converges faster than FP or DO. Instead of relying on best responses, CFR is grounded in the principle of regret minimization. A key result of regret minimization is that, in two-player zero-sum games, the average strategy of two regret minimizing learners converges to a Nash equilibrium. CFR applies this principle by independently minimizing regret at each information state. By appropriately weighting feedback, CFR ensures regret minimization across the entire game, thus guaranteeing convergence to a Nash equilibrium.

Because CFR is more involved than FP and DO, it does not admit the same plug and play workflow with single-agent model-free DRL. Instead, new methods (Steinberger et al., 2020; Gruslys et al., 2020; McAleer et al., 2023) were designed specifically for CFR. Unfortunately, these new methods come with new weaknesses. The most straightforward method (Steinberger et al., 2020), for instance, relies on importance sampling over trajectories, which leads to high variance in feedback, especially in games with many steps. This makes it challenging for function approximators to learn. Although McAleer et al. (2023) resolve this issue, their resolution necessitates a uniform behavioral policy, making it difficult to achieve sufficient coverage of relevant trajectories.

### 2.2.4 Regularization

Motivated by the limitations of FP-, DO-, and CFR-based methods, Perolat et al. (2021) proposed an alternative approach centered around regularization. Unlike previous methods, their approach is designed to converge in the last iterate, aligning it more closely with typical DRL techniques. Perolat et al. (2021)'s work has been influential, serving as inspiration for several subsequent studies (Fu et al., 2022; Perolat et al., 2022; Sokota et al., 2023; Meng et al., 2023; Liu et al., 2024). Among them, Sokota et al. (2023) introduced a policy gradient algorithm called magnet mirror descent (MMD), characterized by the following update rule:

$$\pi_{t+1} = \operatorname*{argmax}_{\pi} \mathbb{E}_{A \sim \pi} q(A) - \alpha \mathrm{KL}(\pi, \rho) - \frac{1}{\eta} \mathrm{KL}(\pi, \pi_t), \tag{1}$$

where $q(A)$ is the value of action $A$, $\alpha$ is a regularization temperature, $\eta$ is a stepsize, $\rho$ is a magnet policy, and KL is KL divergence. In a surprising result, Sokota et al. (2023) demonstrated that MMD, implemented as a standard tabular self-play reinforcement learning algorithm, achieves performance competitive with that of CFR in IIGs. Sokota et al. (2023) also gave empirical evidence that MMD is performant as a DRL algorithm.

## 3 The Policy Gradient Hypothesis

The surprising performance of magnetic mirror descent invites reflection. Consider the functions served by the components of Equation (1):

- $\mathbb{E}_{a \sim \pi} q_t(A)$ maximizes the expected return.
- $-\alpha \mathrm{KL}(\pi, \rho)$ regularizes the updated policy (for uniform $\rho$, this acts similarly to an entropy bonus).
- $-\frac{1}{\eta} \mathrm{KL}(\pi, \pi_t)$ constrains the update size.

Far from being unique to MMD, these functions have long been present in deep PG methods, such as TRPO (Schulman et al., 2015), PPO (Schulman et al., 2017), and PPG (Cobbe et al., 2021). As with MMD, the update rules of these algorithms include terms that maximize expected value and encourage entropy; they also possess mechanisms that, by various means, control the update size.

Yet, despite this shared ethos, other generic PG methods—typically PPO—appear in literature only as underperforming baselines. This raises the question: Why has MMD performed well, while these other generic PG methods have not?

There are multiple answers worth considering. One possibility is that the differences between MMD and other generic PG methods are material. Unlike most PG methods, which use some combination of gradient clipping and forward KL divergence, MMD employs reverse KL divergence to regulate update size. In tabular settings, where competitive performance requires high precision, this distinction is manifestly important. However, in the context of deep learning, it is less clear that this variation matters (Engstrom et al., 2020; Hsu et al., 2020; Chan et al., 2022).

A second possibility is that these generic PG methods are fully capable of performing well, but have not been run with good hyperparameters. Ever-present reasons that baselines tend to perform poorly, such as the expense and difficulty of tuning and the structural disincentives to reporting strong results for baselines, could be at work.

But there are also domain-specific factors that make this second possibility particularly plausible:

- There is a putative belief that generic PG methods do not work in IIGs and that more game theoretically involved algorithms are required. Due to this belief, tuning may have been confounded by confirmation bias.
- Works that report generic PG method baselines often rely on head-to-head results. The intransitive nature of these results increases the complexity of tuning and may thereby have hampered it.
- The hyperparameter regime for which Sokota et al. (2023) showed MMD to be effective involves more entropy regularization than is typically used for PG methods in single-agent problems. Thus, it is possible effective hyperparameter configurations were overlooked due to their regime being too far removed from those of single-agent settings.

Based on these reasons, this work posits that generic PG methods are fully capable of performing well, a position we formalize under the following hypothesis.

> **The Policy Gradient Hypothesis.** *Appropriately tuned policy gradient methods that share an ethos with magnetic mirror descent are competitive with or superior to model-free deep reinforcement learning approaches based on fictitious play, double oracle, or counterfactual regret minimization in two-player zero-sum imperfect-information games.*

The confirmation of the policy gradient hypothesis would have ramifications on both research and practice. On the research side, it would vindicate generic PG methods, which have generally either been dismissed as unsound or relegated to the role of sacrificial baseline. Practically, it would lead to simpler implementations and performance improvements in applications where FP-, DO-, and CFR-based DRL algorithms are currently employed. These applications include autonomous vehicles (Ma et al., 2018), network security (Lu et al., 2020; Xue et al., 2021), robot confrontation (Hu et al., 2022), eavesdropping (Guo et al., 2022), radar (anti-)jamming (Li et al., 2022a;b; Geng et al., 2023; 2024), intrusion response (Hammar and Stadler, 2023), aerial combat (He et al., 2023; Wang et al., 2024), pursuit-evasion games (Li et al., 2023b; 2024b), racing (Zheng et al., 2024), language model alignment (Gemp et al., 2024) and red teaming (Ma et al., 2024).

## 4 BENCHMARKS

Unfortunately, evaluating DRL algorithms for adversarial IIGs is not well standardized. Due to the difficulty of efficiently implementing exploitability for large IIGs, there is a dearth of accessible implementations.

As a result, the literature predominantly resorts to some combination of two unsatisfactory metrics. The first is exploitability in the same small EFGs that served as benchmarks for tabular solvers, such as Leduc Hold'em (Southey et al., 2005) (see Table 4). Benchmarking in these small games is unsatisfactory because success requires learning a precise near-Nash policy for a handful of repeatedly revisited information states—a fundamentally different challenge from large games, where success requires learning a policy that is strong (but not necessarily numerically close to Nash) for a vast number of unvisited information states.

The second is head-to-head evaluations in large games. These evaluations are unsatisfactory because the intransitive dynamics characteristic of imperfect-information games (e.g., rock-beats-scissors-

beats-paper-beats-rock), muddle interpretations of relative strength. Exacerbating the issue, there is no community coordination on head-to-head opponents, precluding direct cross-work comparisons.

This predicament has created an undesirable state of affairs. Many works claim "state-of-the-art" performance, but even experts in specific algorithm classes are often unsure which instances are actually effective. To address this predicament, we set out to implement efficient computations for five large games: 2D5F Liar's Dice (LD2D5F), Phantom Tic-Tac-Toe (PTTT), 3x3 Dark Hex (DH3), Abrupt Phantom Tic-Tac-Toe (APTTT), 3x3 Abrupt Dark Hex (ADH3).

In 2D5F Liar's Dice, each of two players privately rolls two dice with five faces. Players then alternate making claims about the total number of dice across both players that show a given face value (e.g., "two dice showing four"). A legal bid (in the ruleset OpenSpiel adopts) must either increase the quantity or increase the face value while not decreasing the quantity. On a turn, instead of bidding, a player may also call "liar," whereupon all dice are revealed; if the last bid is no greater than the revealed count of the specified face, the the bidder wins, otherwise the caller does.

In Tic-Tac-Toe, which is played on a 3-by-3 grid with square cells, the objective is to get three-in-a-row horizontally, vertically, or diagonally. If the board is filled without either player having three-in-a-row, the game ends in a draw. An example game is shown in Figure 1 (left) in which the first moving player (red) wins via the diagonal.

In 3x3 Hex, which is played on a rhombus with hexagonal cells, the first moving player's objective is to connect the top edge of the board to bottom edge, while the second moving player's objective is to connect the left edge of the board to the right edge. Draws are not possible because the board provably cannot be filled without a player winning (Gale, 1979, Hex Theorem). An example game is shown in Figure 1 (right) in which the first moving player (red) wins by connecting the top and bottom edges.

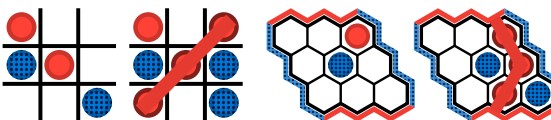

Figure 1: Tic-Tac-Toe (left) and 3x3 Hex (right).

Table 1: Game quantities. Positive Nash values (i.e., expected values for player 1 at Nash equilibria) mean that player 1 has a structural advantage.

| Game | States | Info. states | Nash value |
|---|---|---|---|
| LD2D5F | 235.93M | 15.73M | $0.01770 \pm 1.8e^{-4}$ |
| DH3 | 19.12B | 6.07M | $1.00000 \pm 0.0$ |
| ADH3 | 29.31B | 27.33M | $0.38443 \pm 2.1e^{-4}$ |
| PTTT | 19.93B | 5.99M | $0.66665 \pm 2.6e^{-4}$ |
| APTTT | 27.12B | 23.31M | $0.55017 \pm 1.4e^{-4}$ |

In the *dark* and *phantom* variants of these games, actions are concealed from the non-acting player, creating imperfect information. These imperfect-information variants have two rulesets—classical and *abrupt*—which differ in how they handle the situation in which the acting player attempts to place a piece on a cell occupied by their opponent's piece. In classical, the acting player must select a different cell. In abrupt, the acting player loses their turn.

These games have at least three significant strengths as benchmarks:

1. OpenSpiel (Lanctot et al., 2019) support: LD2D5F, DH3, PTTT, ADH3 are already implemented in OpenSpiel, making it easy for other researchers to use them. To address APTTT's absence, we provide our own game implementation.

2. Precedent: Existing work uses LD, PTTT, and ADH3 for head-to-head evaluations (Sokota et al., 2023; McAleer et al., 2023; Meng et al., 2023; Liu et al., 2024), providing external evidence of their relevance.

3. Size: These games allow model-free DRL algorithms to be trained in minutes or hours on commodity hardware. But, as detailed in Table 1, they have millions of infostates, making them quite large by the standards of game solving benchmarks.

Indeed, the size of these games makes it nontrivial to implement exploitability computations. To accommodate their magnitude, we use sequence-form representations, which are approximately multiple orders of magnitude more compact than the explicit game trees, alongside dynamic payoff matrix generation to circumvent explicit matrix storage. This memory-efficient approach enables usage on commodity hardware.

To optimize the runtime of LD2D5F, we exploit the facts that actions are public and chance outcomes are independent and located at the root of the game to use dynamic programming to calculate the payoff at all public states in linear time, similarly to Brown et al. (2020). To optimize the runtime of the 3x3 games, we distribute computation across multiple threads after the first two moves, using 18 buffers to prevent concurrency conflicts across the 81 possible opening sequences. Depending on the machine and game, this implementation delivers exploitability computation (and head-to-head evaluation) times of roughly 30 to 90 seconds.

## 5 EXPERIMENTS

While our high-performance exploitability computations open the door to evaluating model-free DRL for IIGs more rigorously, the novelty of these benchmarks necessitates particular care in experimental design and interpretation. Simply showing that tuned PG methods perform "well" on these benchmarks would not suffice as evidence for the policy gradient hypothesis, as there are no numbers for FP-, DO-, or CFR-based DRL algorithms against which to compare. The obvious recourse—which we take—is to report our own results for FP, DO, and CFR-based DRL algorithms. However, since we have hypothesized these algorithms will underperform, our role in tuning them should be carefully scrutinized. To aid readers in this endeavor, we provide detailed documentation of our algorithm implementations, hyperparameter tuning procedure, and hyperparameter tuning results, in addition to final results.

### 5.1 ALGORITHMS AND IMPLEMENTATION

We choose algorithms to evaluate using two criteria. First, that the collective set of algorithms includes representatives from FP-based algorithms, DO-based algorithms, CFR-based algorithms, and generic PG methods, so as to make our experiments sufficiently comprehensive to provide evidence for or against the policy gradient hypothesis. Second, that algorithms be implemented by a reliable external source in a fashion requiring as little adaptation as possible for OpenSpiel (Lanctot et al., 2019) compatibility, so as to reduce the probability of an implementation error.

In accordance with these criteria, the algorithms we select are NFSP (Heinrich and Silver, 2016), PSRO (Lanctot et al., 2017), ESCHER (McAleer et al., 2023), R-NaD (Perolat et al., 2022), PPO (Schulman et al., 2017), PPG (Cobbe et al., 2021), and MMD (Sokota et al., 2023). This selection includes one FP-based algorithm (NFSP), one DO-based algorithm (PSRO), one CFR-based algorithm (ESCHER), three generic PG methods (PPO, PPG, MMD), and one non-standard PG method (R-NaD) inspired by Perolat et al. (2021). Algorithm implementations were sourced from Open-Spiel (Lanctot et al., 2019), except PPG and ESCHER, which we sourced from CleanRL (Huang et al., 2022) and the official ESCHER repository (McAleer et al., 2023), respectively.

Of the algorithm implementation code that we wrote or modified, some of the larger potential sources of error include moving NFSP from TensorFlow (Abadi et al., 2015) to PyTorch (Paszke et al., 2019), adding multi-agent support for PPO and PPG, and implementing MMD as a modification of PPO. To corroborate the correctness of these pieces of code, we reproduce the NFSP exploitability results for Leduc Hold'em from OpenSpiel, as detailed in Appendix D.1, and the MMD approximate exploitability results for ADH3 and PTTT from Sokota et al. (2023), as detailed in Appendix D.2. We enable external verification of the correctness of these pieces of code, and the rest of our algorithm implementations, by open sourcing our codebase at `github.com/nathanlct/IIG-RL-Benchmark`.

### 5.2 DESIGN

We design our training runs as a series of two launches: a hyperparameter tuning launch followed by a maximization-bias-free evaluation launch.

For the hyperparameter tuning launch, for each combination of the 5 games and 7 algorithms, we test 50 hyperparameter configurations for 10 million steps over 3 seeds. To create these configurations, we modify the default positive real-valued hyperparameters by independent randomly sampled powers of 2—multiplying for the vast majority of hyperparameters, but exponentiating for some hyperparameters upper bounded by 1. We take the default values for hyperparameters from

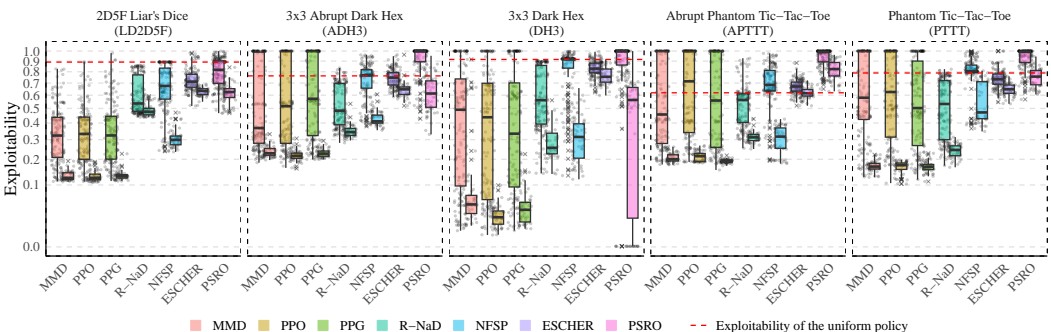

Figure 2: **Exploitability results.** For each combination of game and algorithm, the box-and-whisker pair depicts the distribution of final exploitability over the runs from the hyperparameter tuning launch (left) and evaluation launch (right) with square-root y-axis scale. R-NaD, NFSP, ESCHER, and PSRO failed to outperform generic PG methods (MMD, PPO, PPG).

the implementation source in almost all cases. The most notable exceptions to this are the network architecture and optimizer, for which we impose a 3-layer 512-hidden unit fully connected network and Adam (Kingma and Ba, 2015) on every algorithm, and the entropy coefficient, for which we apply the value from Sokota et al. (2023) to all generic PG methods. We provide a comprehensive list of hyperparameters, default values, and links to the specific lines of code of the external sources from which those default values were taken in Appendix G.

For the evaluation launch, we run the 5 strongest hyperparameter configurations from the first launch (measured by lowest average final exploitability), with 10 fresh seeds, for 10 million steps, for each combination of the 5 games and 7 algorithms. From these evaluation runs, we report both exploitabilities and head-to-head comparisons. For the head-to-head comparisons, we select the policy with the lower median final exploitability over the 10 seeds for each hyperparameter configuration, as well as an approximate Nash equilibrium policy computed using discounted CFR (Brown and Sandholm, 2019a), and compare these selected policies for each pair of algorithms for each game.

## 5.3 RESULTS

We summarize two of the main results: exploitability performance and head-to-head performance.

**Exploitability** Figure 2 plots the distribution of final exploitabilities for each game and algorithm in pairs of box-and-whisker plots for the hyperparameter tuning launch (left) and the evaluation launch (right). Over these runs, ESCHER was uniformly noncompetitive. PSRO was largely noncompetitive, showing poor performance in ADH3, PTTT, and APTTT and having only sporadic success in DH3, for which several hyperparameter configurations found Nash equilibria on some seeds but not others. NFSP and R-NaD approached or matched the performance of generic PG methods in select cases, but typically underperformed them. The generic PG methods performed the strongest, and were roughly on par with one another.

**Head-to-Head Comparisons** Figure 3 plots head-to-head comparisons for each pair of algorithms. Relative performance patterns are similar to those for exploitability: The generic PG methods matched or defeated other algorithms, while maintaining rough parity among themselves; R-NaD either approached the performance of the generic PG methods or underperformed them, while NFSP and, especially, ESCHER and PSRO were noncompetitive.

## 6 DISCUSSION

The experimental results unambiguously support the policy gradient hypothesis. Not only did the FP-, DO-, and CFR-based DRL algorithms fail to outperform the generic PG methods, they, alongside the nonstandard PG method, were largely noncompetitive.

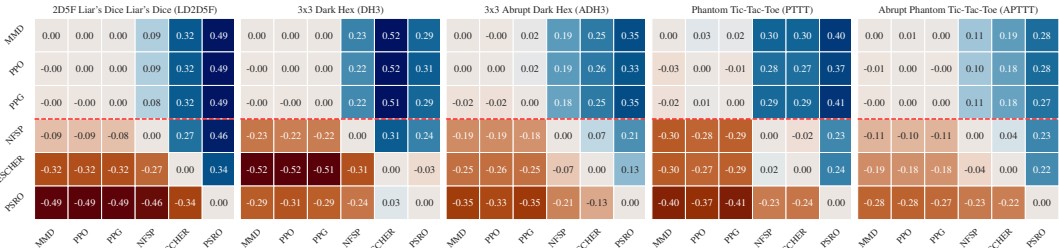

Figure 3: **Head-to-head evaluations.** The number in each cell is the expected return of the row algorithm against the column algorithm when each plays half of the games as the first moving player. R-NaD, NFSP, ESCHER, and PSRO failed to outperform generic PG methods (MMD, PPO, PPG), which are segregated by the dashed red lines.

However, we emphasize that we make no claim—and should not be interpreted as having claimed—that we have established as true the policy gradient hypothesis or any other speculation discussed in this work. A prudent interpretation of our experiments gives consideration to multiple limitations.

First, we only performed experiments on five games, four of which are somewhat homogeneous. We do not believe that the relative performance of the algorithms we evaluated would be universally consistent across adversarial IIGs at large (indeed, it is not even consistent across these five games).

Second, we evaluated only one FP-based algorithm (the original NFSP), one DO-based algorithm (the original PSRO), and one CFR-based algorithm (the most recent adaption to DRL). As discussed in Section 2.2, there are dozens of variants of these algorithms, especially PSRO. Any one of these variants may possess important innovations that result in improved performance on our benchmarks.

Third, the experimental results reflect our particular experimental conditions, including: 1) the default hyperparameter values, 2) the hyperparameters we tuned, 3) the manner in which we tuned hyperparameters, and 4) performance evaluation at 10 million steps. These conditions are indeed particular, and should not be misconstrued to represent optimal algorithm performance. It is possible that strong performing hyperparameters exist for NFSP, PSRO, ESCHER, or R-NaD, but were not found by our hyperparameter sweep because they are too many orders of magnitude away from the defaults, or because they require too many different defaults to be changed simultaneously, or because they involve different categorical hyperparameters—which we did not tune. It is also possible that the hyperparameters we used would achieve strong performance, given more training steps—a possibility consistent with preliminary experiments, which suggested that all algorithms continue to decrease exploitability beyond 10 million steps.

To power community efforts to address or investigate these second and third sets of limitations, as well as to give creators of existing and future DRL algorithms for IIGs the opportunity to demonstrate the efficacy of their algorithms, we release our exploitability computations in a Python package with OpenSpiel-compatible bindings at `github.com/gabrfarina/exp-a-spiel`. We detail this package, called `exp-a-spiel`, in Appendix A. We encourage its use not only as it pertains to the policy gradient hypothesis, but also more generally for future IIGs research.

## 7 CONCLUSION

This work hypothesizes and presents evidence that generic deep PG methods are strong approaches for adversarial IIGs. Our results, along with those of Yu et al. (2022), who demonstrated PPO's effectiveness in cooperative IIGs, and Sokota et al. (2023), who demonstrated MMD's effectiveness in adversarial IIGs, add to a growing body of evidence for the possibility that a single, simple PG method could serve as a universal algorithm for DRL in games. We believe this possibility is an aspiration worth pursuing.

## 8 ACKNOWLEDGMENTS

This work was supported by Award #2125858 NRT-AI: Convergent, Responsible, and Ethical Artificial Intelligence Training Experience for Roboticists; NSF #2340651; NSF #2402650; DARPA #HR00112490431; ARO W911NF-24-1-0193; the NYU IT High-Performance Computing resources, services, and staff expertise; and ONR grant #N000142212121.

We thank Jeremy Cohen and Alexander Robey for helpful feedback.

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

# A  THE EXP-A-SPIEL PACKAGE

We release a Python library to efficiently compute exploitability of policies for the five benchmark games introduced in Section 4. The library supports outputting state representations compatible with OpenSpiel (Lanctot et al., 2019), making it straightforward to evaluate the exploitability of policies trained using the latter library.

## A.1  EXPLOITABILITY COMPUTATION

Internally, exp-a-spiel evaluates the exploitability of policies by carrying out computations using the so-called *sequence-form* representation of the five games (Romanovskii, 1962; von Stengel, 1996; Koller et al., 1996). This representation is roughly 1000 times more compact than the actual game trees, allowing us to compute exact exploitability (and head-to-head values) on commodity hardware. This representation is comprised of two objects: (i) the *treeplexes* $\mathcal{X}, \mathcal{Y}$ of the two players (Hoda et al., 2010), and (ii) the sequence-form payoff matrix $A$ of the game. Every policy for player 1 admits an equivalent vector $x \in \mathcal{X}$ (resp. $y \in \mathcal{Y}$ for player 2), and the expected payoff corresponding to player 1 can be computed in closed form via the bilinear form $x^\top A y$. Due to the size of the games, the sequence-form payoff matrix $A$ is not stored explicitly in memory by exp-a-spiel, but is rather materialized on the fly in a multi-threaded fashion as needed.

At a high level, given two policies $\pi_1, \pi_2$ for the players, exp-a-spiel computes exploitability by performing the following.

1. First, $\pi_1$ and $\pi_2$ are converted into their sequence-form equivalents $x \in \mathcal{X}, y \in \mathcal{Y}$. This step requires memory and runtime that scale linearly in the number of information states of the games. The latter is in the order of $10^7$ as shown in Table 1.
2. Then, the vectors $g_1 \coloneqq Ay$, $g_2 \coloneqq -A^\top x$ are computed. These are the gradients of the utility function of the game. We accelerate the computation by playing the first two actions of the game and then each thread calculates the gradient assuming the first two moves separately and then the gradients are safely reduced. It is worth noting that while there are 81 possible pairs of first moves for the players, only 18 buffers are required to avoid all possible concurrency conflicts. Depending on the machine and the game, calculating the gradients take roughly between 30 and 90 seconds. The compute time for calculating best response on various games is presented in Table 2.
3. Finally, exploitability is computed according to the formula

$$\max_{\hat{x} \in \mathcal{X}} \hat{x}^\top g_1 + \max_{\hat{y} \in \mathcal{Y}} y^\top g_2.$$

   The optimization problems can be solved in closed form using memory and runtime linear in the number of information states of the games, by using a standard greedy algorithm on treeplexes.

## A.2  STATE-OF-THE-ART TABULAR SOLVERS

exp-a-spiel also includes implementation of the following state-of-the-art tabular solvers:

1. CFR+ (Tammelin, 2014) substitutes the regret matching (RM) algorithm in CFR with regret matching+ (RM+), leading to better performance empirically. The main difference between RM and RM+ is that RM+ clamps the regrets to be non-negative.
2. Discounted CFR (DCFR) (Brown and Sandholm, 2019b), in a similar vain as CFR+, discounts the regrets to reduce the effect of previous actions with large (absolute) regret. Practically speaking, at iteration $t$, we rescale the negative regrets by $1 - 1/(1 + t^\beta)$ and the positive regrets by $1 - 1/(1 + t^\alpha)$. We use the default value of $\alpha = 1.5$ and $\beta = 0$.
3. Predictive CFR (PCFR) (Farina et al., 2019) uses optimistic regret minimizers to achieve empirical and theoretical faster convergence rates for CFR. Practically, we observe the gradients twice to calculate the behavior policy of CFR.

These are online, first-order optimization algorithms that tabularly refine strategies—represented in sequence-form—by taking steps in the direction of the gradient utility. Table 3 reports the exploitability of equilibrium computed by the methods across the 3x3 games and Figure 4 shows the

Table 2: Runtime for computing best response with `exp-a-spiel` and OpenSpiel in seconds. 'OOM' indicates that the process ran out of memory before timing out. All results in this table were run on an AMD EPYC 7413 24-Core Processor with 256 GB of memory.

| Game | OpenSpiel | exp-a-spiel |
|---|---|---|
| Liar's Dice 1d4f | 1 | 0.03 |
| Liar's Dice 1d5f | 1 | 0.03 |
| Liar's Dice 1d6f | 5 | 0.04 |
| Liar's Dice 2d3f | 10 | 0.04 |
| Liar's Dice 2d4f | 527 | 0.45 |
| Liar's Dice 2d5f | OOM | 11 |
| Classical PTTT | OOM | 27 |
| Abrupt PTTT | OOM | 27 |
| Classical DH3 | OOM | 47 |
| Abrupt DH3 | OOM | 47 |

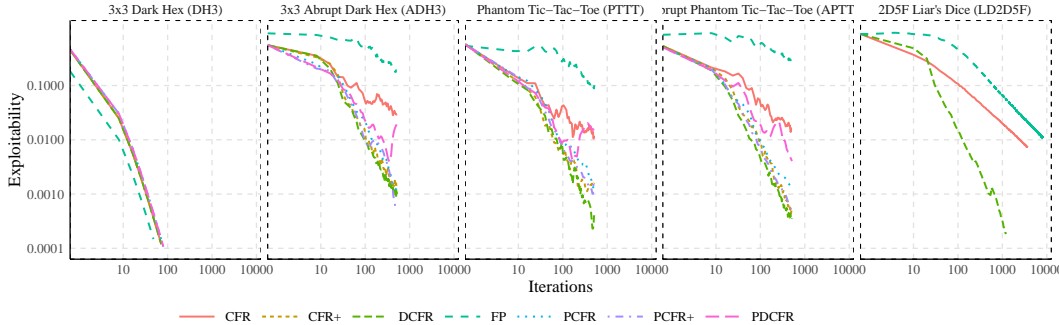

Figure 4: Performance of various tabular methods.

Table 3: Exploitability after 512 iterations of different algorithms on 3x3 board games

| Game | Algorithm | Value | Exploitability |
|---|---|---|---|
| 3x3 Dark Hex (DH3) | CFR | 0.9999993 | 0.0000004 |
| | CFR+ | 0.9999993 | 0.0000004 |
| | DCFR | 0.9999993 | 0.0000004 |
| | FP | 0.9999997 | 0.0000002 |
| | PCFR | 0.9999993 | 0.0000005 |
| | PCFR+ | 0.9999993 | 0.0000005 |
| | PDCFR | 0.9999993 | 0.0000005 |
| 3x3 Abrupt Dark Hex (ADH3) | CFR | 0.3841795 | 0.0295619 |
| | CFR+ | 0.3844054 | 0.0011629 |
| | DCFR | 0.3844321 | 0.0010063 |
| | FP | 0.3580483 | 0.1929647 |
| | PCFR | 0.3844107 | 0.0011789 |
| | PCFR+ | 0.3843936 | 0.0006123 |
| | PDCFR | 0.3844073 | 0.0192829 |
| Phantom Tic-Tac-Toe (PTTT) | CFR | 0.6664844 | 0.0107796 |
| | CFR+ | 0.6666324 | 0.0016106 |
| | DCFR | 0.6666511 | 0.0004070 |
| | FP | 0.6622969 | 0.1016475 |
| | PCFR | 0.6660216 | 0.0011638 |
| | PCFR+ | 0.6662418 | 0.0019337 |
| | PDCFR | 0.6662910 | 0.0157007 |
| Abrupt Phantom Tic-Tac-Toe (APTTT) | CFR | 0.5501850 | 0.0162306 |
| | CFR+ | 0.5501825 | 0.0004936 |
| | DCFR | 0.5501728 | 0.0003577 |
| | FP | 0.5428041 | 0.2871734 |
| | PCFR | 0.5501960 | 0.0011708 |
| | PCFR+ | 0.5501886 | 0.0004846 |
| | PDCFR | 0.5502121 | 0.0040427 |

convergence of DCFR and PCFR+. We note that Dark Hex has a winning deterministic strategy for the first moving player. We analyze this strategy in Appendix B.

## B  SOLVING DARK HEX 3

We investigate a deterministic winning strategy for the first moving player in Dark Hex 3. The value of this policy against a uniform policy is 1, implying that it wins against all possible deterministic strategies, including any possible best response to it. The strategy is shown in Figure 5 where, at each information state, a list of actions to try in that order is given. If the first action fails, the next action is played, and so on. After playing an action, there are two possible outcomes: the action is playable, or information is gained. In general, the size of the list of actions is equal to one plus the number of opponent pieces that have been played but have not been observed; thus, the last action has to be playable. Beyond the computational proof that the strategy in Figure 5 is optimal, we

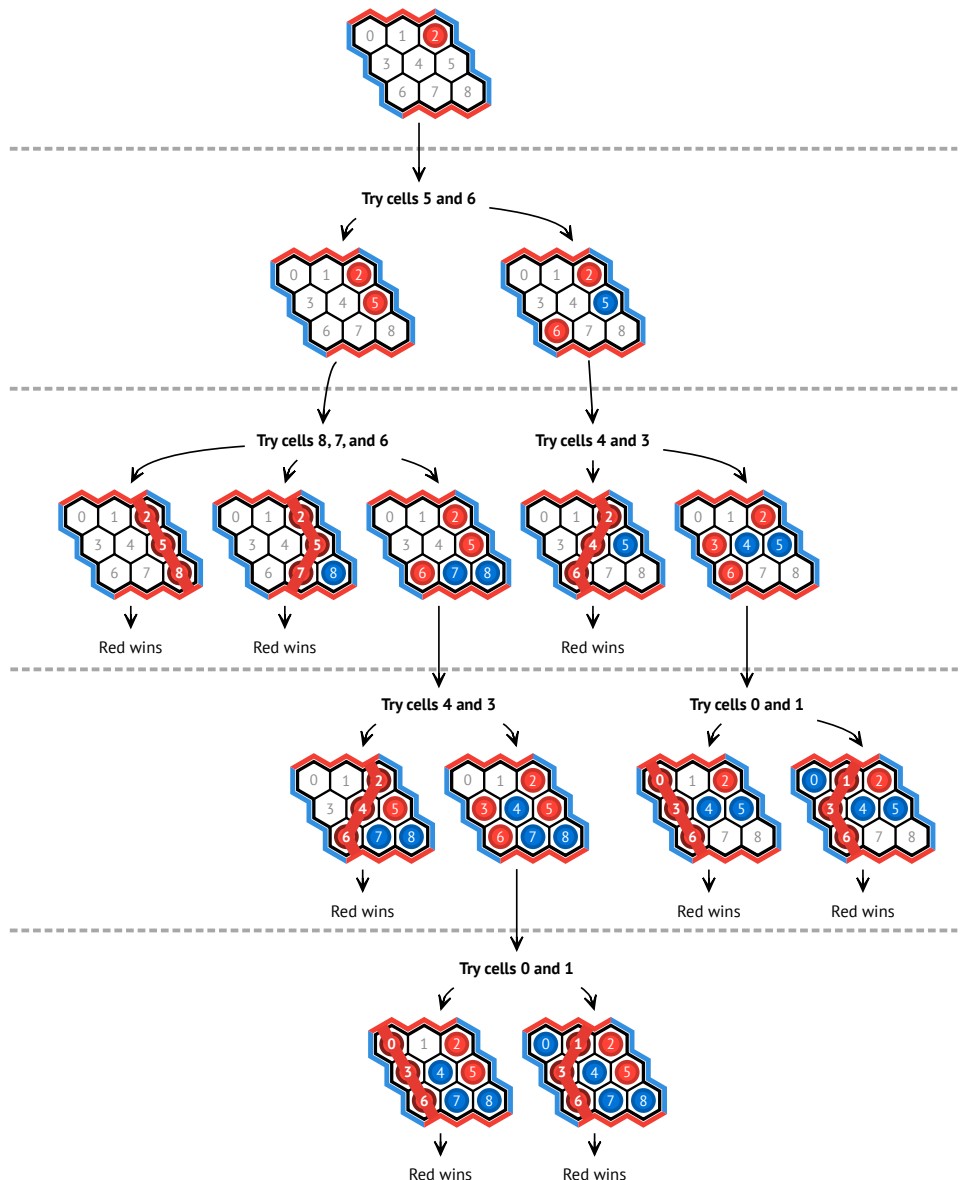

Figure 5: A deterministic strategy for player 1 that always wins. The gray dashed lines denote a hidden action of the blue player.

can show it using the fact the game of Hex (with perfect information) cannot end in a draw (Gale, 1979, Hex Theorem). Since the first player always wins in the reachable information states, it is impossible the invisible pieces are in a way that the opponent has already won.

This strategy is not applicable for Abrupt 3x3 Dark Hex as the board in the abrupt version does not correspond to a board of hex. For instance, player 2 might never put a piece on the board.

Since the game of Hex can be won by the first player, we argue that any deterministic winning Nash equilibrium for Dark Hex must also be an optimal strategy in the perfect information game, otherwise there must exist a deterministic strategy for the opponent that wins against this Nash equilibrium. Thus at any information state, the action played must be either invalid or a winning move for all states that are compatible with that information state (i.e., it is possible that they are the state of the board at that information state). This gives rise to an algorithm for finding deterministic Nash equilibrium by backtracking over which action to play. We suspect that this algorithm does not

have any solution beyond 3 by 3 Hex and that Dark Hex 4 does not have any winning deterministic strategy.

## C  ALGORITHM IMPLEMENTATION DETAILS

We considered the following algorithms in this work: NFSP, PSRO, R-NaD, ESCHER, PPO, PPG and MMD. Below we give details about the implementation of each of them.

**NFSP**  (Source Implementation): We use the implementation of NFSP in OpenSpiel (Lanctot et al., 2019). This implementation learns distinct models for player 1 and player 2. For consistency with the other algorithms, we rewrote some of the algorithm to use PyTorch (Paszke et al., 2019) instead of TensorFlow (Abadi et al., 2015).

**PSRO**  (Source Implementation): We use the implementation of PSRO in OpenSpiel (Lanctot et al., 2019). This implementation learns distinct models for player 1 and player 2. For consistency with the other algorithms, we replaced the default OpenSpiel Tensorflow DQN agent (source) with the OpenSpiel PyTorch DQN agent (source).

**R-NaD**  (Source Implementation): We use the implementation of R-NaD in OpenSpiel (Lanctot et al., 2019). This implementation learns a single model that is used for both player 1 and player 2. Natively, this algorithm is written in JAX (Bradbury et al., 2018). For convenience during evaluation, we convert the neural networks learned in JAX to equivalent PyTorch models. We verified this conversion by ensuring that, for the same inputs, the outputs of the models are equivalent.

**ESCHER**  (Source Implementation): We use the implementation of ESCHER from the original paper (McAleer et al., 2023). This implementation learns a single policy network that is used for both player 1 and player 2. Natively this algorithm is written in Tensorflow. For convenience during evaluation we convert the neural networks learned in Tensorflow to equivalent PyTorch models and verify by ensuring each layer weight and bias is equivalent.

**PPO**  (Source Implementation) We use the implementation of PPO in OpenSpiel, which is itself a modification from the CleanRL PPO made to work with OpenSpiel games and legal action masking. We further modify it to support self-play, and we learn a single model for both players.

**PPG**  (Source Implementation) We use the implementation of PPG in CleanRL, which we modify to make it work with OpenSpiel games, legal action masking and to support self-play. We learn a single model for both players.

**MMD**  We use our PPO implementation and simply add a backward KL term in the PPO loss. We learn a single model for both players.

## D  RESULTS SUPPORTING IMPLEMENTATION CORRECTNESS

### D.1  NFSP

We verified this re-implementation by comparing the exploitability of strategies learned in the original file with ours on Leduc Hold'em, see Figure 6.

### D.2  MMD

We verified that we obtain results consistent with the original MMD implementation (Sokota et al., 2023). Namely, after a 10M steps training we obtain an approximate exploitability of around 0.14 and 0.20 on PTTT and ADH3 respectively. For reference, Table 2 in Sokota et al. (2023) reports an approximate exploitability of $0.15 \pm 0.01$ and $0.20 \pm 0.01$ for PTTT and ADH3 respectively.

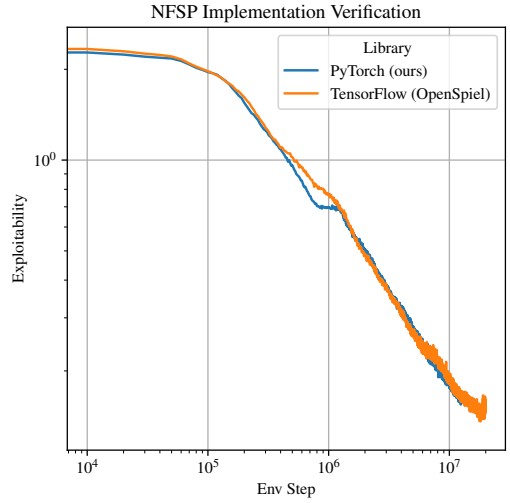

Figure 6: Exploitability performance of the TensorFlow NFSP implementation in OpenSpiel and our PyTorch adaptation on Leduc Hold'em aggregated over 2 seeds.

Table 4: Related work and their use of exact or approximate exploitability when evaluating their method.

| Paper | Exact Exploitability | Approx. Exploitability or Head-to-Head |
|---|---|---|
| NFSP (Heinrich and Silver, 2016) | Leduc | |
| PSRO (Lanctot et al., 2017) | Leduc | |
| ESCHER (McAleer et al., 2023) | 2x2 Battle Ship!, Leduc | Dark Chess, DH5, PTTT |
| RNaD (Perolat et al., 2022) | | Stratego |
| MMD (Sokota et al., 2023) | LD1D4F, Leduc, ADH2 | ADH3, PTTT |
| Qin et al. (2019) | Kuhn Poker | |
| AdaptFSP (Goldstein and Brown, 2023) | Leduc | |
| Hu et al. (2023) | Leduc | |
| NSFP-PLT (Li et al., 2023a) | | 6-, 3-, 2-player poker, Leduc |
| Pipeline PSRO (McAleer et al., 2020) | Leduc | Barrage |
| XDO (McAleer et al., 2021) | Leduc | |
| Smith et al. (2021) | Leduc | |
| Anytime PSRO(McAleer et al., 2022a) | | Leduc |
| Online DO (Dinh et al., 2022) | | |
| Zhou et al. (2022) | | Leduc |
| PDO (Yao et al., 2023) | Leduc | |
| SP-PSRO (McAleer et al., 2022b) | LD1D6F, 2x2 Battle Ship!, Leduc | |
| Self-Adaptive PSRO (Li et al., 2024a) | LD1D6F, Leduc | |
| Fusion PSRO (Lian, 2024) | LD1D6F, Leduc | |
| Dream (Steinberger et al., 2020) | Leduc | Flop Hold'em |
| ARMAC (Gruslys et al., 2020) | LD1D6F, Leduc | Head's up |
| FoReL (Perolat et al., 2021) | LD1D6F, Leduc | |
| Meng et al. (2023) | Leduc | PTTT, DH5, DH4 |
| Liu et al. (2024) | Leduc | |

Table 5: Size of games for which exact exploitability is reported in prior work.

| Game | Number of Infosets | Number of States (including terminal) |
|---|---|---|
| ADH2 | 94 | 471 |
| Leduc | 936 | 9457 |
| LD1D4F | 1024 | 8181 |
| 2x2 Battle Ship! | 3286 | 10069 |
| LD1D6F | 24576 | 294883 |

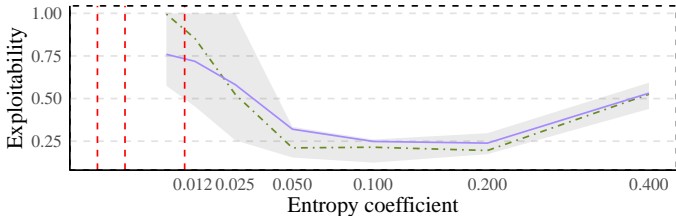

Figure 7: Average (dash-dotted green) and median (solid purple) exploitability of generic policy gradient methods across all games as a function of entropy coefficient, with shaded interquartile range and square-root x-axis. Vertical dashed red lines show the default entropy coefficient values for PPO in widely used DRL libraries. Results broken down by game are shown in Figure 12.

Table 6: Entropy coefficients used in popular reinforcement learning libraries for policy gradient algorithms.

| Library | Entropy Coefficient |
|---|---|
| Stable Baselines (Hill et al., 2018) | 0.0 |
| CleanRL (Huang et al., 2022) | 0.01 or 0.001 |
| RLlib (Liang et al., 2018) | 0.0 |
| OpenSpiel (Lanctot et al., 2019) | 0.01 |
| PufferLib (Suarez, 2024) | 0.01 |
| RL-Games (Makoviichuk and Makoviychuk, 2021) | 0 or 0.01 |
| Tianshou (Weng et al., 2022) | 0 or 0.01 |

# E  ENTROPY COEFFICIENT ANALYSIS

In Section 3, we suggest that one possible reason for the poor performance of generic PG methods in existing literature is that the entropy coefficients required to achieve good performance are too far removed from typical choices. To investigate, we plot exploitability as a function of entropy coefficient in Figure 7. The results are consistent with this possible explanation: Not only do unilateral changes in entropy coefficient produce drastic changes in exploitability, but the entropy coefficients with the best average performance are between 0.05 and 0.2, larger than any of the default entropy coefficients for PPO in Stable Baselines (Hill et al., 2018), CleanRL (Huang et al., 2022), RLlib (Liang et al., 2018), OpenSpiel (Lanctot et al., 2019), PufferLib (Suarez, 2024), RL-Games (Makoviichuk and Makoviychuk, 2021), and Tianshou (Weng et al., 2022), which, as detailed in Table 6, range between 0 and 0.01.

# F  ADDITIONAL RESULTS

The evolution of exploitability over 10M training steps for all 5250 hyperparameter tuning runs is shown in Figure 8. Each algorithm-game pair, includes 150 runs—50 hyperparameter sets with 3 seeds each. The variance across these 3 seeds per set is visualized in Figure 9. Furthermore, we compute the approximate importance of each hyperparameter in the tuning process, which we report in Figure 10. Given the high importance we obtained for the entropy coefficient in the policy-gradient algorithms, in Figure 12 we specifically analyze the average exploitability as a function of the entropy coefficient. We observe that on average we obtain the best exploitability results for entropy coefficients much higher than standard ones detailed in Table 6. We also performed some additional experiments on a wider hyperparameter sampling range for the `number_training_episodes` hyperparameter in PSRO, shown in Table 7. Figure 11 reports the training durations and memory usage of the evaluated algorithms. Note that these metrics depend on hardware and implementation optimizations and are provided for reference only.

Table 7: This table shows the performance of PSRO on our benchmark games using a wider sampling range for the `number_training_episodes` hyperparameter that determines for how long each oracle trains. Exploitability is presented as percentiles across 50 different hyperparameter samples. Results for the original sampling range presented in the main results of the paper ($[125, 8000]$) are shown for comparison

| Game | Sampling Range | expl. @ 10% | expl. @ 25% | expl. @ 50% | expl. @ 75% |
|------|----------------|-------------|-------------|-------------|-------------|
| ADH3 | $[2500, 160000]$ | 0.929014 | 1 | 1 | 1 |
| ADH3 | $[125, 8000]$ | 0.750014 | 0.894206 | 1 | 1 |
| APTTT | $[2500, 160000]$ | 0.934627 | 1 | 1 | 1 |
| APTTT | $[125, 8000]$ | 0.795704 | 0.893034 | 0.988603 | 1 |
| DH3 | $[2500, 160000]$ | 0.969553 | 1 | 1 | 1 |
| DH3 | $[125, 8000]$ | 0.72258 | 0.862278 | 1 | 1 |
| PTTT | $[2500, 160000]$ | 0.88345 | 1 | 1 | 1 |
| PTTT | $[125, 8000]$ | 0.786203 | 0.886317 | 0.99254 | 1 |

We show the training exploitability curves for the 1750 runs of the evaluation launch in Figure 13. The variance across seeds for each hyperparameter set can be seen in Figure 9. Additionally, Figure 15 reports the training exploitability curves for the best hyperparameter set of each algorithm-game pair as well as the variance over 10 seeds for each set.

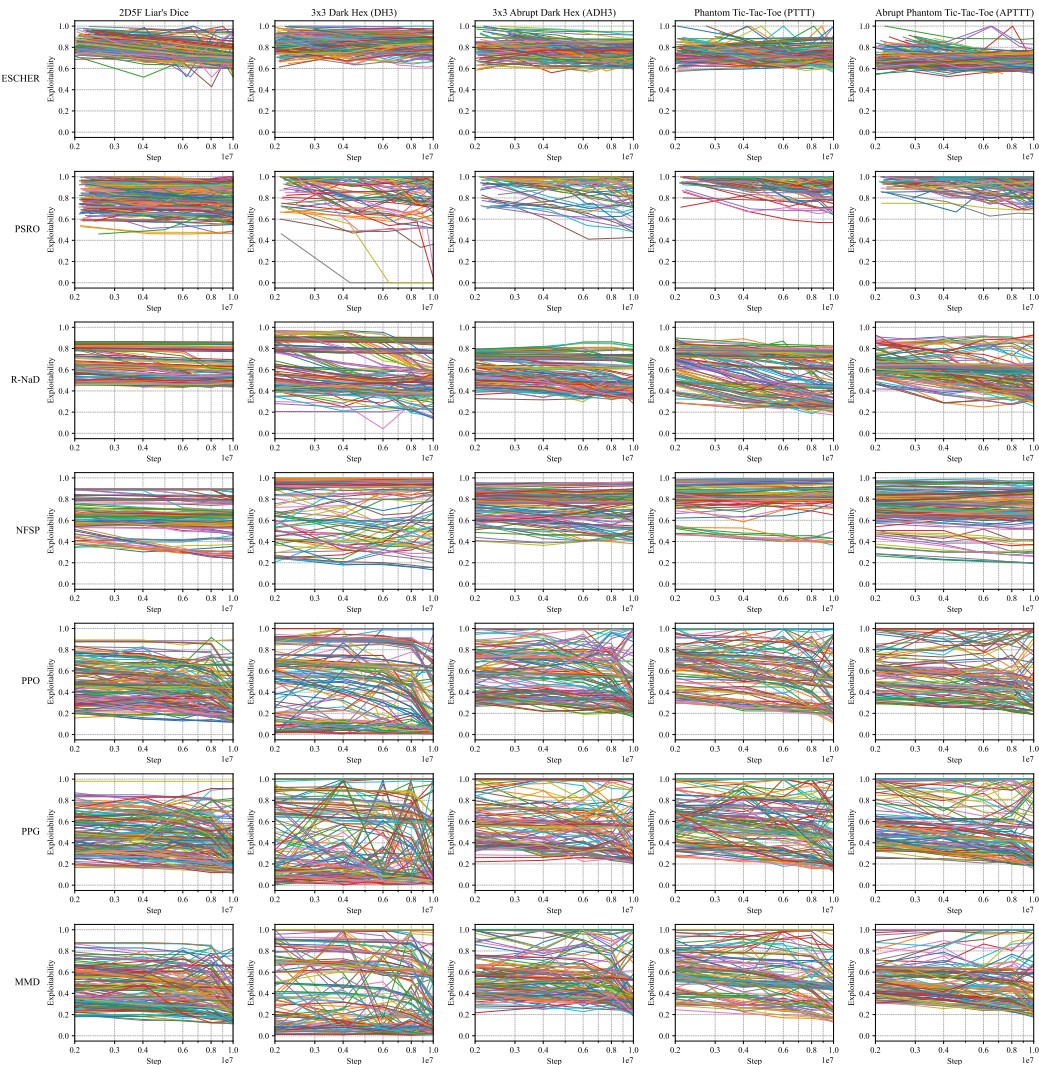

Figure 8: Exploitability vs. step for all 5250 runs of the hyperparameter tuning launch, broken down by game and algorithm.

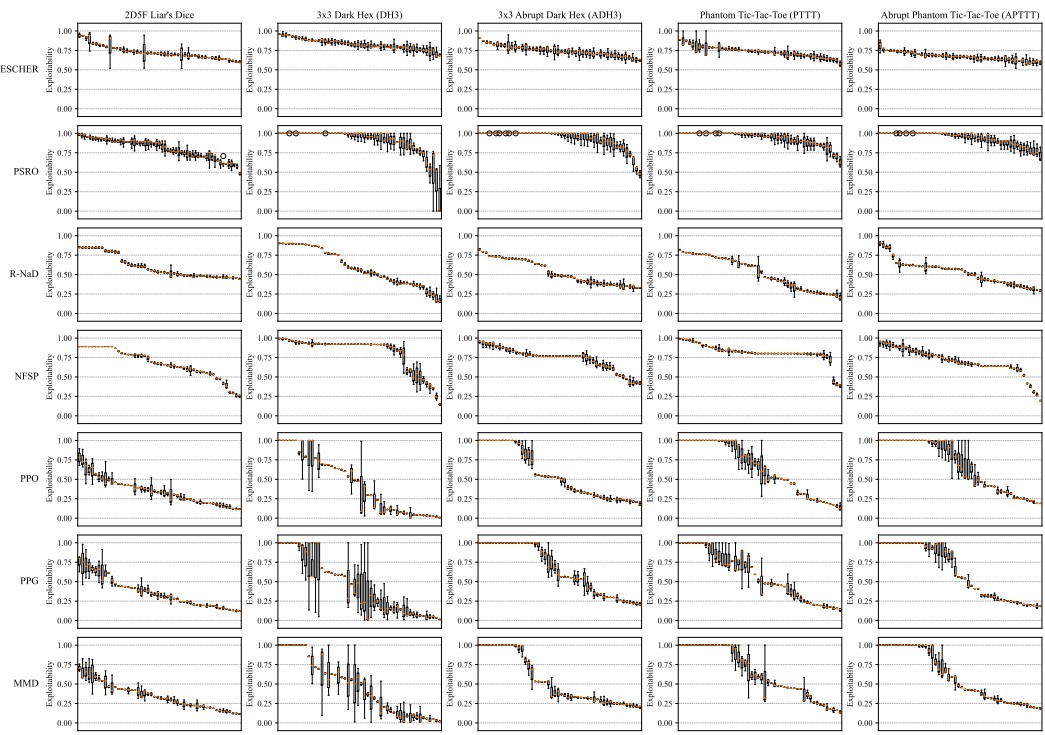

Figure 9: Exploitability for all 5250 runs of the hyperparameter tuning launch, broken down by game and algorithm, then grouped by set of hyperparameters. The boxes-and-whiskers each show the variance over 3 seeds for one set of hyperparameters, and are ordered by decreasing average exploitability.

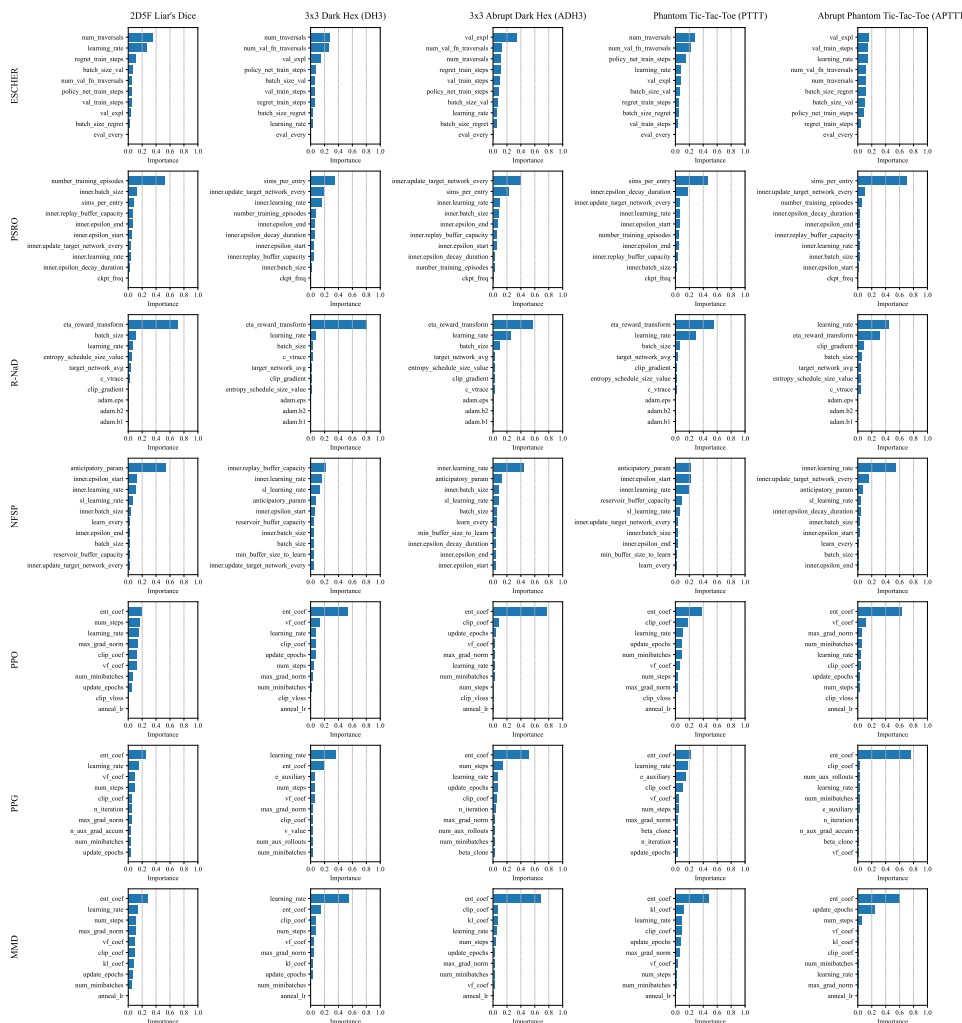

Figure 10: The top 10 most influencial hyperparameters from the hyperparameter tuning launch, broken down by game and algorithm. The sweep results provide mappings from hyperparameter sets to exploitabilities, which we use to train a random forest regression model. This model assigns each hyperparameter a coefficient representing its importance in the prediction. This importance reflects the hyperparameter's impact on exploitability: a high value indicates a strong influence, while a low value suggests minimal impact. Note: inner_rl_agent was abbreviated to `inner`.

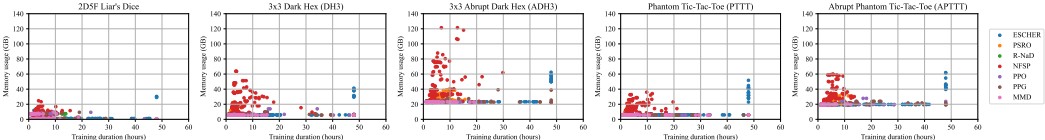

Figure 11: Memory usage (RAM) and wall-time training duration statistics from the hyperparameter tuning launch, broken down by games and algorithm.

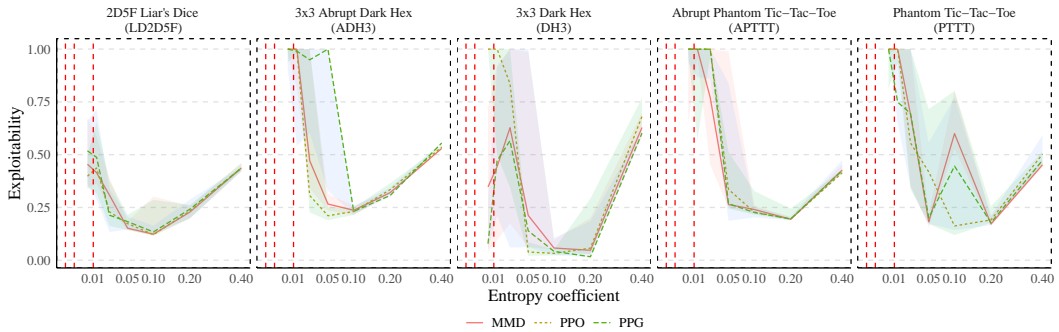

Figure 12: Figure 7 separated on a per game and algorithm basis.

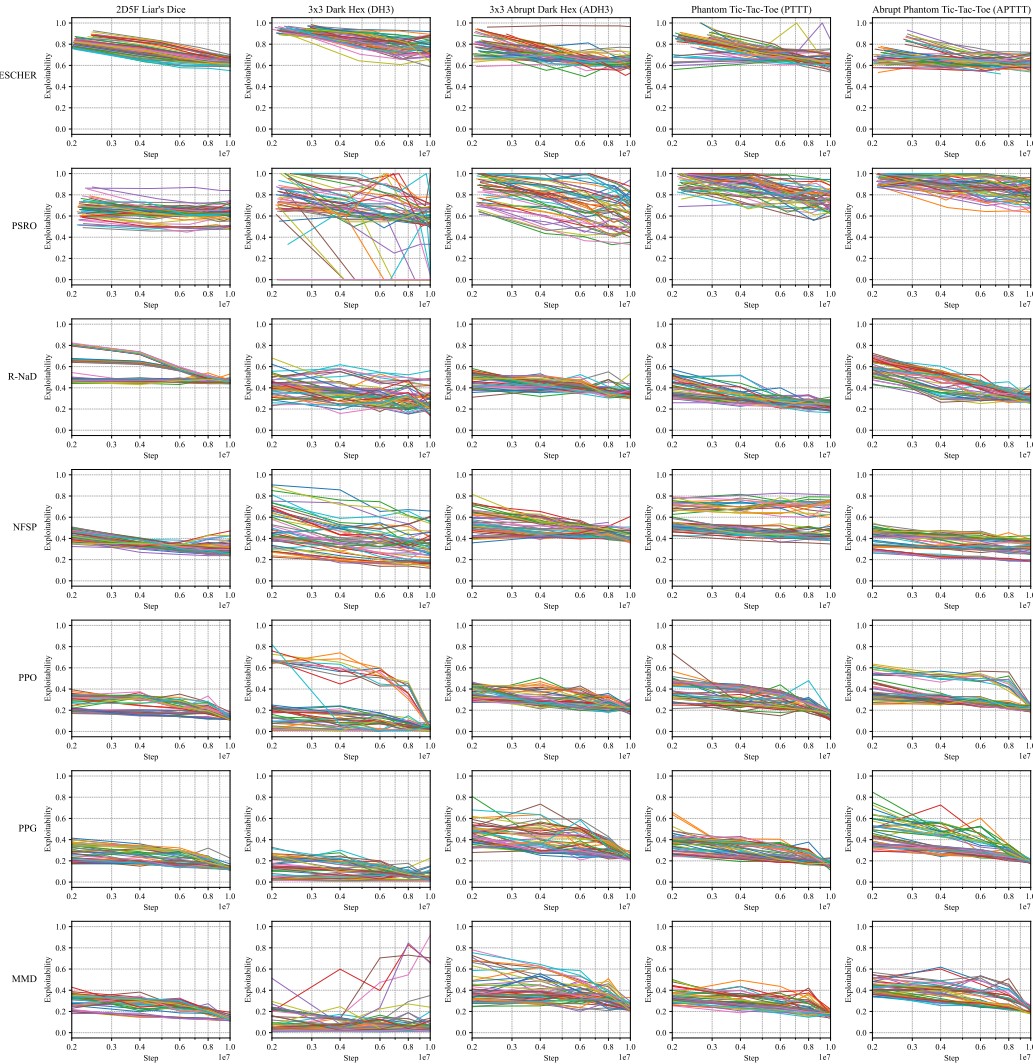

Figure 13: Exploitability vs. step for all 1750 runs of the evaluation launch, broken down by game and algorithm.

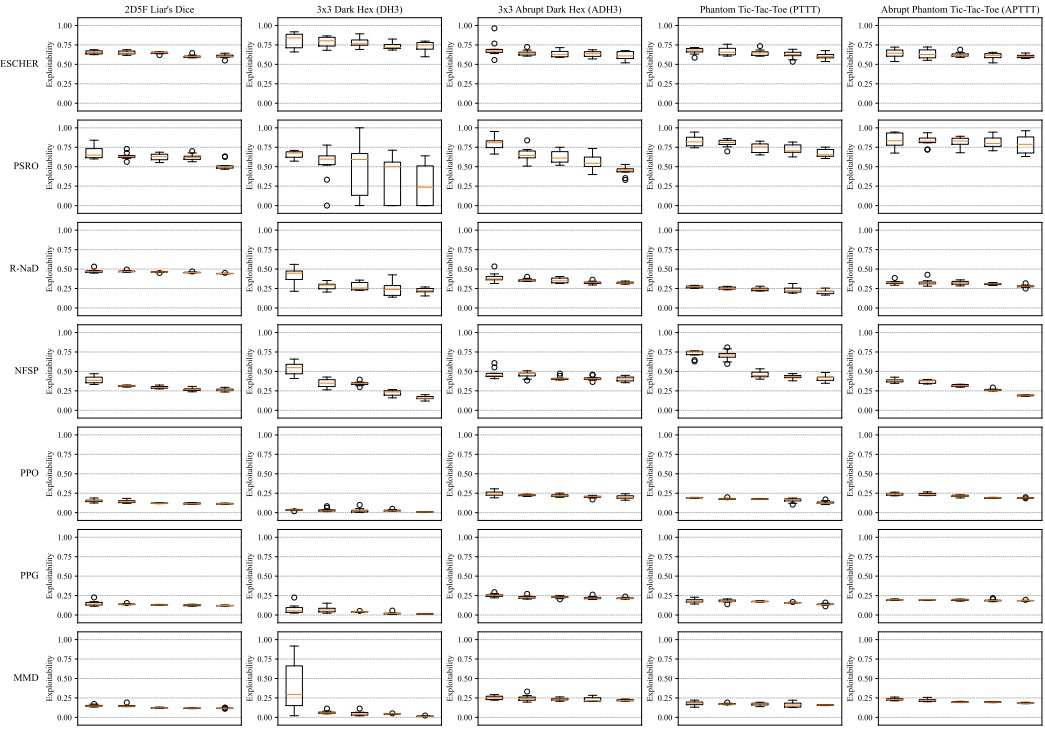

Figure 14: Exploitability for all 1750 runs of the evaluation launch, broken down by game and algorithm, then grouped by set of hyperparameters. The boxes-and-whiskers each show the variance over 10 seeds for one set of hyperparameters, and are ordered by decreasing average exploitability.

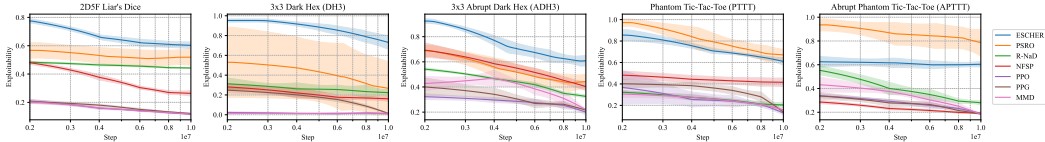

Figure 15: Variance across seeds for the best-performing hyperparameter set in the evaluation launch, broken down by game and algorithm. The best set is defined as having the smallest final exploitability, averaged over seeds. Plots show mean exploitability over 10M training steps (solid line) with a shaded region representing the standard deviation range across 10 seeds.

# G ALGORITHM HYPERPARAMETERS

Here, we provide an overview of the hyperparameters for the algorithms employed in the experiments. For each hyperparameter being swept, for each run, we sample $N \sim$ Uniform$(\{-3, -2, -1, 0, 1, 2, 3\})$. For positive real-valued hyperparameters, we adjust the hyperparameter by multiplying its default value by $2^N$, rounding to an integer when necessary. For hyperparameters that may otherwise egress $[0, 1]$ when they should not, we exponentiate the default value to the power of $2^N$, rather than multiplying. All algorithms were trained for 10 million steps and with the same neural network architecture (3 fully connected layers of 512 neurons each).

## G.1 NFSP

**Tuned hyperparameters:**

1. `replay_buffer_capacity`
   - Description: Size of DQN replay buffer.
   - Default value: $2 \times 10^5$
   - Source: OpenSpiel NFSP Leduc example.
   - Maximum value: $5 \times 10^5$, imposed to ensure that the replay buffer capacity cannot be significantly larger than the reservoir buffer capacity, in alignment with the ratio of default values.

2. `reservoir_buffer_capacity`
   - Description: Size of reservoir buffer.
   - Default value: $2 \times 10^6$
   - Source: OpenSpiel NFSP Leduc example.
   - Maximum value: $4 \times 10^6$, imposed due to memory constraints.

3. `min_buffer_size_to_learn`
   - Minimum number of instances in buffer before training.
   - Default value: 1000
   - Source: OpenSpiel NFSP Leduc example.

4. `anticipatory_param`
   - Description: Probability of using the RL best response as policy.
   - Default value: 0.1
   - Source: OpenSpiel NFSP Leduc example.

5. `batch_size`
   - Description: Number of transitions to sample at each learning step.
   - Default value: 128
   - Source: OpenSpiel NFSP Leduc example.

6. `learn_every`
   - Description: Number of environment steps between learning updates.
   - Default value: 64
   - Source: OpenSpiel NFSP Leduc example.

7. `sl_learning_rate`
   - Description: Learning rate for supervised learning network.
   - Default value: 0.01
   - Source: OpenSpiel NFSP Leduc example.

8. `inner_rl_agent.epsilon_decay_duration`
   - Description: Number of game steps over which exploration decays.
   - Default value: $1 \times 10^7$
   - Source: OpenSpiel NFSP Leduc example.

- The epsilon decay duration is in the Leduc Example is set to decay completely throughout training ($2 \times 10^7$ games). We mimic this behavior by setting the decay duration to the total number of training steps ($1 \times 10^7$ steps).

9. **inner_rl_agent.learning_rate**
   - Description: Learning rate for Q-network.
   - Default value: 0.01
   - Source: OpenSpiel NFSP Leduc example.

10. **inner_rl_agent.batch_size**
    - Description: Batch size of Q-network.
    - Default value: 128
    - Source: OpenSpiel NFSP Leduc example.

11. **inner_rl_agent.update_target_network_every**
    - Description: Number of steps between target network updates.
    - Default value: 19200
    - Source: OpenSpiel NFSP Leduc Example

12. **inner_rl_agent.epsilon_start**
    - Description: Initial exploration value.
    - Default value: 0.06
    - Source: OpenSpiel NFSP Leduc example.

13. **inner_rl_agent.epsilon_end**
    - Description: Final exploration value.
    - Default value: 0.001
    - Source: OpenSpiel NFSP Leduc example.

**Fixed hyperparameters:**

1. **optimizer_str**
   - Description: Supervised learning network optimizer.
   - Value: Adam

2. **loss_str**
   - Description: Loss function for Q-network.
   - Value: MSE

### G.2 PSRO

**Tuned Hyperparameters**

1. **sims_per_entry**
   - Description: Number of games to play for estimating the meta game.
   - Value: 1000
   - Source: OpenSpiel PSRO example.

2. **number_training_episodes**,
   - Description: Number of episodes over which to train each oracle.
   - Value: 1000
   - Source: OpenSpiel PSRO RL oracle file.

3. **inner_rl_agent.epsilon_decay_duration**
   - Description: Number of game steps over which exploration decays.
   - Default value: $1 \times 10^7$
   - Source: OpenSpiel NFSP Leduc example.

- Note: the epsilon decay duration in the Leduc Example is set to decay completely throughout training ($2 \times 10^7$ games). We mimic this behavior by setting the decay duration to the total number of training steps ($1 \times 10^7$ steps).

4. **inner_rl_agent.learning_rate**

   - Description: Learning rate for Q-network.
   - Default value: 0.01
   - Source: OpenSpiel PSRO v2 example.

5. **inner_rl_agent.batch_size**

   - Description: Batch size for Q-network.
   - Default value: 128
   - Source: OpenSpiel DQN Default

6. **inner_rl_agent.update_target_network_every**

   - Description: Number of steps between target network updates.
   - Default value: 1000
   - Source: OpenSpiel PSRO v2 Example

7. **inner_rl_agent.epsilon_start**

   - Description: Initial exploration value.
   - Default value: 0.06
   - Source: OpenSpiel NFSP Leduc example.

8. **inner_rl_agent.epsilon_end**

   - Description: Final exploration value.
   - Default value: 0.001
   - Source: OpenSpiel NFSP Leduc example.

**Fixed Hyperparameters**

1. **optimizer_str**

   - Description: Oracle agent learning network optimizer
   - Value: Adam

2. **loss_str**

   - Description: Loss function for Q-network
   - Value: MSE

3. **training_strategy_selector**

   - Description: Determines against which oracle agents to train in the next iteration
   - Value: Probabilistic (distributed according to the meta strategy)

## G.3    R-NAD

**Tuned hyperparameters:**

1. **batch_size**

   - Description: Number of samples in each training batch.
   - Default value: 256
   - Source: OpenSpiel R-NaD defaults.

2. **learning_rate**

   - Description: Learning rate for the optimizer.
   - Default value: $5 \times 10^{-5}$
   - Source: OpenSpiel R-NaD defaults.

3. **clip_gradient**

- Description: Maximum gradient value for clipping.
- Default value: 10,000
- Source: OpenSpiel R-NaD defaults.

4. **target_network_avg**
   - Description: Smoothing factor for target network updates.
   - Default value: 0.001
   - Source: OpenSpiel R-NaD defaults.

5. **eta_reward_transform**
   - Description: Scaling factor for regularization in reward transformation.
   - Default value: 0.2
   - Source: OpenSpiel R-NaD defaults.

6. **entropy_schedule_size_value**
   - Description: Schedule for updating the regularization network.
   - Default value: 50,000
   - Source: OpenSpiel R-NaD Leduc defaults.

7. **c_vtrace**
   - Description: Coefficient for V-trace importance weights (Espeholt et al., 2018).
   - Default value: 1.0
   - Source: OpenSpiel R-NaD defaults.

**Fixed hyperparameters:**

1. **trajectory_max**
   - Description: Number of steps after which games are truncated.
   - Value: Disabled

2. **beta**
   - Description: Size of the gradient clipped threshold in the NeurD gradient clipping (Hennes et al., 2019).
   - Default value: 2.0
   - Source: OpenSpiel R-NaD defaults.

3. **clip**
   - Description: Size of clipping for the importance sampling in the NeurD gradient clipping (Hennes et al., 2019).
   - Default value: 10,000
   - Source: OpenSpiel R-NaD defaults.

### G.4   ESCHER

**Tuned hyperparameters**

1. **num_traversals**
   - Description: Number of game plays for regret function learning.
   - Default value: 1,000
   - Source: Hyperparameters for Phantom TTT and Dark Hex 4 from McAleer et al. (2023).

2. **num_val_fn_traversals**
   - Description: Number of game plays for value function learning.
   - Default value: 1,000
   - Source: Hyperparameters for Phantom TTT and Dark Hex 4 from McAleer et al. (2023).

3. **regret_train_steps**
   - Description: Number of training steps for regret network.
   - Default value: 5,000
   - Source: Hyperparameters for Phantom TTT and Dark Hex 4 from McAleer et al. (2023).

4. **val_train_steps**
   - Description: Number of training steps for value function network.
   - Default value: 5,000
   - Source: Hyperparameters for Phantom TTT and Dark Hex 4 from McAleer et al. (2023).

5. **policy_net_train_steps**
   - Description: Number of training steps for the policy network.
   - Default value: 10,000
   - Source: Hyperparameters for Phantom TTT and Dark Hex 4 from McAleer et al. (2023).

6. **batch_size_regret**
   - Description: Batch size for regret network learning.
   - Default value: 2,048
   - Source: Hyperparameters for Phantom TTT and Dark Hex 4 from McAleer et al. (2023).

7. **batch_size_val**
   - Description: Batch size for value function network.
   - Default value: 2,048
   - Source: Hyperparameters for Phantom TTT and Dark Hex 4 from McAleer et al. (2023).

8. **learning_rate**
   - Description: Gradient descent learning rate.
   - Default value: $1 \times 10^{-3}$
   - Source: ESCHER Codebase.
   - Notes: No value found in paper.

9. **val_expl**
   - Description: Uniform policy mixing rate for off-policy exploration for value network.
   - Default value: 0.01
   - Source: ESCHER Codebase.
   - Notes: No value found in paper.

**Fixed hyperparameters:** We re-use all the fixed hyperparameters from the ESCHER Codebase. As there are many, we do not list them here.

### G.5 PPO

**Tuned hyperparameters**

1. **learning_rate**
   - Description: Optimizer learning rate.
   - Default value: $2.5 \times 10^{-4}$
   - Source: OpenSpiel's PPO Implementation.

2. **num_steps**
   - Description: The number of steps to run in each environment per policy rollout (i.e. the batch size is num_steps x num_envs).

- Default value: 128
- Source: OpenSpiel's PPO Example Implementation.

3. **num_minibatches**

   - Description: The number of minibatches (i.e. the minibatch size is `round(num_steps x num_envs / num_minibatches)`.
   - Default value: 4
   - Source: OpenSpiel's PPO Example Implementation.

4. **update_epochs**

   - Description: Number of policy update epochs (i.e. how many times to go through the whole batch in each iteration).
   - Default value: 4
   - Source: OpenSpiel's PPO Example Implementation.

5. **clip_coef**

   - Description: Clipping coefficient $\epsilon$ in the PPO loss.
   - Default value: 0.1
   - Source: OpenSpiel's PPO Example Implementation.

6. **ent_coef**

   - Description: Coefficient for the entropy bonus in the loss.
   - Default value: 0.05
   - Source: Inspired from (Sokota et al., 2023).
   - Note: This value is larger than in OpenSpiel's PPO Implementation because we have found that entropy bonuses lead to much better policies. However, the default value of 0.01 is still within our sweeping range.

7. **vf_coef**

   - Description: Coefficient for the value term in the loss.
   - Default value: 0.5
   - Source: OpenSpiel's PPO Example Implementation.

8. **max_grad_norm**

   - Description: Maximum norm of the gradient allowed during gradient clipping.
   - Default value: 0.5
   - Source: OpenSpiel's PPO Example Implementation.

**Fixed hyperparameters**

1. **num_envs**

   - Description: The number of parallel game environments.
   - Default value: 8
   - Source: OpenSpiel's PPO Example Implementation.

2. **anneal_lr**

   - Description: Toggle for learning rate annealing for the policy and value networks.
   - Default value: `True`
   - Source: OpenSpiel's PPO Example Implementation

3. **gamma**

   - Description: Discount factor $\gamma$ for the return.
   - Default value: 0.99
   - Source: OpenSpiel's PPO Example Implementation.
   - Note: We do not sweep this parameter due to the short-horizon nature of the games.

4. **gae_lambda**

   - Description: Coefficient $\lambda$ for the general advantage estimation (GAE).

- Default value: 0.95
- Source: OpenSpiel's PPO Example Implementation.
- Note: We do not sweep this parameter due to the short-horizon nature of the games.

5. **norm_adv**
   - Description: Toggle for advantage normalization before computing the loss.
   - Default value: `True`
   - Source: OpenSpiel's PPO Example Implementation.

6. **clip_vloss**
   - Description: Whether or not to clip the values in the value loss computation.
   - Default value: `True`
   - Source: OpenSpiel's PPO Example Implementation.

7. **target_kl**
   - Description: Target KL divergence threshold for early stopping of training epochs.
   - Default value: `None` (disabled)
   - Source: OpenSpiel's PPO Example Implementation.

## G.6 PPG

We consider the same parameters as in PPO (Section G.5), with the addition of the following ones:

**Tuned hyperparameters**

1. **n_iteration**
   - Description: Number of policy updates in the policy phase ($N_\pi$).
   - Default value: 32
   - Source: CleanRL implementation and Cobbe et al. (2021).

2. **e_policy**
   - Description: Number of policy updates in the policy phase ($E_\pi$).
   - Default value: 1
   - Source: CleanRL implementation and Cobbe et al. (2021).

3. **v_value**
   - Description: Number of value updates in the policy phase ($E_V$).
   - Default value: 1
   - Source: CleanRL implementation and Cobbe et al. (2021).

4. **e_auxiliary**
   - Description: Number of epochs to update in the auxiliary phase ($E_{\text{aux}}$).
   - Default value: 6
   - Source: CleanRL implementation and Cobbe et al. (2021).

5. **beta_clone**
   - Description: Behavior cloning coefficient.
   - Default value: 1.0
   - Source: CleanRL implementation and Cobbe et al. (2021).

6. **num_aux_rollouts**
   - Description: Number of mini-batches in the auxiliary phase.
   - Default value: 4
   - Source: CleanRL implementation.

7. **n_aux_grad_accum**
   - Description: Number of gradient accumulations in each mini-batch.
   - Default value: 1
   - Source: CleanRL implementation.

### G.7 MMD

We consider the same parameters as in PPO (Section G.5), with the addition of the following ones:

**Tuned hyperparameters**

1. `kl_coef`
   - Description: Coefficient of the reverse KL divergence in the loss function.
   - Default value: 0.05
   - Source: (Sokota et al., 2023).
   - Note: We use a constant value for the reverse KL coefficient (as well as for the entropy coefficient) instead of a custom schedule.

