# OpenReview forum: "Reevaluating Policy Gradient Methods for Imperfect-Information Games"
_ICLR.cc/2026/Conference — ICLR 2026 Poster_

### Official Review · Reviewer_WVeg · 2025-10-29

**Soundness:** 2
**Presentation:** 2
**Contribution:** 3
**Rating:** 4
**Confidence:** 4

**Summary:**

The paper presents empirical evidence that commonly used policy gradient algorithms, such as PPO and PPG, which have some similarity to MMD, quickly converge to low exploitability strategies in a range of imperfect information zero-sum games. Moreover, they show that within a restricted budget, they converge much closer to the Nash equilibrium, than selected methods based on CFR, Fictitious play, and Double Oracle algorithms.

**Strengths:**

* The paper presents interesting and important empirical evidence that may encourage people to try a simpler and more standard solution to solve their problems.
  * The paper makes a rigorous effort to compare to several baselines.
  * The paper promises to publish a more efficient BR computation for OpenSpiel, which might be useful

**Weaknesses:**

Summary: I believe the empirical results are interesting and likely worth publishing. However, the writing of the paper is not ideal. The authors try to condense the main message of the paper to a short paragraph they call a "hypothesis". However, for me, a hypothesis is a formal statement that can be proven to be true or false. The "hypothesis" in this paper is a very vague observation that is subject to many interpretations and many of them are clearly not true in general.

Moreover, there are some mistakes, likely caused by insufficient proofreading after adapting the paper to the new template, and several misleading statements. I would be happy to recommend accepting the paper after a major revision, but in the current state, I lean towards rejecting.


Detailed comments:

"Hypothesis" is IMO not the right word for what the paper provides. I would be fine if it is called an "observation" and softened by some "tend to" "often", or something similar. If it were to stay a "hypothesis", it would have to specify much more clearly what is "share an ethos with magnetic mirror descent" and list specific classes of methods "based on FP, DO, CFR", since it is a huge diverse set of algorithms.

I also do not like the authors over-selling the contribution of implementing efficient best response, calling it "first broadly accessible exact exploitability computation" and not properly reviewing prior efforts to scale-up BR computation, or comparing it to the BR computation available in other frameworks. I would expect mentioning at least [A], arguing why an approximate solution, such as [B] is not sufficient, and comparing execution speed to at least OpenSpiel BR, if not also some other frameworks. Otherwise, it is hard to accept that as a contribution.

The paper IMO falsely claims that "only a small number of works make the effort to report exploitability in large games". A very large portion of papers on approximating equilibria in the last over a decade use exploitability as the key metric and report it on large poker endgames, Phantom TicTacToe itself, and other games of comparable size. It is certainly nice to publish a new, efficient implementation compatible with OpenSpiel and presumably much faster than the default implementation, but I do not consider similar claims to be an appropriate characterisation of the existing work.

Moreover, the paper should establish that one or two orders of magnitude that their BR implementation may buy compared to the existing ones is crucial to deliver their results. Would similar experiments on slightly smaller games deliver substantially different results? It claims that it is "a fundamentally different challenge", but gives a very brief statement to support that and does not show or specify whether it becomes fundamentally different at 10^8, 10^9 or 10^10 states.

I also consider misleading the claim that "DO can suffer from slow convergence (...) according to the (...)  exponential lower bound for DO", after talking about perfect recall EFGs up to that point in the paper. The boud they cite is not about this class of games. The paper explicitly states in the introduction that there are efficient variants of DO for EFGs.


To conclude, I would love to see this paper published, with more fair treatment of the existing work, more exact definition of "the policy gradient observation", and experiments on smaller variants of the games showing whether it starts to hold only form some scale of the games up, or whether it holds also for smaller games. Moreover, I would avoid implying that the methods are better than all CFR, DO, and FP based methods, and focus on using these methods as mere baselines that are outperformed.


[A] Johanson, Michael, et al. "Accelerating best response calculation in large extensive games." IJCAI. Vol. 11. 2011.

[B] Timbers, Finbarr, et al. "Approximate Exploitability: Learning a Best Response."

**Questions:**

Q1)  Do PPO/PPG match the performance of MMD even on smaller games?

Q2) Do you know of specific counterexamples where PPO/PPG fail to converge?

Q3) Is the plot in Figure 2 "the hyperparameter tuning launch" or the "evaluation launch"?

Q4) Do I understand correctly that you run PSRO with at most 8000 steps per oracle training?

Q5) Can you provide a compute time comparison between your BR implementations and the OpenSpiel one?

---

> ### Author Response · Authors · 2025-11-19
> **Author response (1/3)**
>
> > "Hypothesis" is IMO not the right word for what the paper provides. I would be fine if it is called an "observation" and softened by some "tend to" "often", or something similar. If it were to stay a "hypothesis", it would have to specify much more clearly what is "share an ethos with magnetic mirror descent" and list specific classes of methods "based on FP, DO, CFR", since it is a huge diverse set of algorithms.
>
> We are happy to consider using a term different than hypothesis if the reviewer feels that it would be more appropriate. However, "observation" would not work because the hypothesis preceded the experiments (and therefore the observation) and we are hypothesizing it to hold more generally (not merely observing that it held in this case).
>
> > it would have to specify much more clearly what is "share an ethos with magnetic mirror descent"
>
> By share an ethos we mean perform the functions discussed in Section 3, namely, maximizing the return, regularizing the update, and constraining the size of the update.
>
> > and list specific classes of methods "based on FP, DO, CFR", since it is a huge diverse set of algorithms.
>
> We mean the entire set of algorithms---the hypothesis is a strong one.
>
> > The paper IMO falsely claims that "only a small number of works make the effort to report exploitability in large games". A very large portion of papers on approximating equilibria in the last over a decade use exploitability as the key metric and report it on large poker endgames, Phantom TicTacToe itself, and other games of comparable size.
>
> As is clear from the context of the quotation, this claim is specific to the literature of model-free deep reinforcement learning algorithms for imperfect-information games. This claim is indeed true, as we evidence below.
>
> Here is a list of all papers in the literature that we cited, along with the largest games for which they reported exact exploitability:
>
> |Paper|Games with exact exploitability
> |-----|-------------------
> |NFSP|Leduc |
> |PSRO|Leduc|
> |ESCHER| 2x2 Battle Ship!, Leduc|
> |RNaD| None|
> |MMD|  1d4f LD, Leduc, ADH2|
> |[1]| Kuhn Poker|
> |AdaptFSP|Leduc|
> |[3]|Leduc|
> |NFSP-PLT|None
> |Pipeline PSRO| Leduc|
> |XDO|Leduc|
> |[5]| Leduc|
> |Anytime PSRO | None|
> |Online DO| None|
> |[2] | None|
> |[6] PDO | Leduc |
> |SP-PSRO | 1d6f LD, 2x2 Battle Ship, Leduc|
> |Self-adaptive PSRO|  1d6f LD, Leduc|
> |Fusion PSRO| 1d6f LD, Leduc|
> |Dream| Leduc|
> |The Advantage Regret-Matching Actor-Critic|1d6f LD, Leduc|
> |FoReL | 1d6f LD, Leduc|
> |[4] | Leduc|
> |[8] | Leduc
>
>
> Here are the sizes of those games:
> |Game|Number of Infosets| Number of states (including terminal)|
> |----|-----|---|
> |ADH2| 94 | 471
> |Leduc| 936 | 9457 |
> |1d4f Liar's Dice| 1024 | 8181|
> | 2x2 Battle Ship| 3286 | 10069|
> |1d6f Liar's Dice| 24576 | 294883|
>
>
> None of these games are large relative to those for which we report exact exploitability.
>
> Here are the largest games for which head-to-head evaluations or approximate exploitability is reported.
> |Paper|Games with RL approximate exploitability or head-to-head evals|
> |-----|-------------------|
> |NFSP|
> |PSRO| |
> |ESCHER| Dark Chess,DH5, PTTT|
> |RNaD|  Stratego|
> |MMD|  ADH3, PTTT |
> |[1]|
> |AdaptFSP|
> |[3]|
> |NFSP-PLT| 6, 3, 2 player poker, Leduc, |
> |Pipeline PSRO| Barrage|
> |XDO|
> |[5]|
> |Anytime PSRO  | Leduc|
> |Online DO|
> |[2] |  Leduc
> |[6] PDO
> |SP-PSRO
> |Self-adaptive PSRO|
> |Fusion PSRO|
> |Dream| Flop Hold'em
> |The Advantage Regret-Matching Actor-Critic| Head's up|
> |FoReL |
> |[4] |PTTT, Dh5, Dh4|
> |[8] |
>
> As you can see, only 3 papers use PTTT:
> - ESCHER
> - MMD
> - [4]
>
> and none of them report exact exploitability.
>
> [1] Improving Fictitious Play Reinforcement Learning with Expanding Models
> [2] Efficient Policy Space Response Oracles
> [3] Fictitious Self-Play Reinforcement Learning with Expanding Value Estimation
> [4] An Efficient Deep Reinforcement Learning Algorithm for Solving Imperfect Information Extensive-Form Games
> [5]Iterative Empirical Game Solving via Single Policy Best Response
> [6] Policy Space Diversity for Non-Transitive Games
> [7] Regret-Minimizing Double Oracle for Extensive-Form Games
> [8] Regularization is Enough For Last-Iterate Convergence in Zero-Sum Games
>
> > It is certainly nice to publish a new, efficient implementation compatible with OpenSpiel and presumably much faster than the default implementation, but I do not consider similar claims to be an appropriate characterisation of the existing work.
>
> We believe there is a misunderstanding here. The issue we are addressing is not that OpenSpiel's "default implementation" is slow. It's that, as a matter of practice, OpenSpiel's "default implementation" does not support these games. See the table at the end of the rebuttal.

---

> > ### Author Response · Authors · 2025-11-19
> > **Author response (2/3)**
> >
> > > I also do not like the authors over-selling the contribution of implementing efficient best response, calling it "first broadly accessible exact exploitability computation" and not properly reviewing prior efforts to scale-up BR computation, or comparing it to the BR computation available in other frameworks. I would expect mentioning at least [A], arguing why an approximate solution, such as [B] is not sufficient, and comparing execution speed to at least OpenSpiel BR, if not also some other frameworks. Otherwise, it is hard to accept that as a contribution.
> >
> > We are in complete agreement that there have been other efforts to scale up exploitability computations (such as [A]) and that such efforts have scaled to games much larger than those in the submission. However, we stand by the characterization that these are the first broadly accessible implementations. (Certainly, none are sufficiently accessible that previous work in this literature felt inclined to reported exploitability using them, as shown by the table above.)
> >
> > Regarding approximate exploitability [B], the reasons that it is less desirable include that it is slower, inconsistent across different works, and inaccurate (it is not uncommon to see "negative" approximate exploitabilities reported. eg, https://openreview.net/forum?id=07o2iouN1Y).
> >
> >
> > > I also consider misleading the claim that "DO can suffer from slow convergence (...) according to the (...) exponential lower bound for DO", after talking about perfect recall EFGs up to that point in the paper. The boud they cite is not about this class of games. The paper explicitly states in the introduction that there are efficient variants of DO for EFGs.
> >
> > Thanks for pointing this out. We were referring to the POSG result but agree that this was confusing given our EFG-adjacent notation. We removed the sentence.
> >
> > > Would similar experiments on slightly smaller games deliver substantially different results?
> >
> > We show results for such an experiment below. The experiments are over 50 hyperparameter configurations with 1 seed each for each algorithm/game pair. The small games we chose are instances of Liar's Dice for which OpenSpiel is suited to calculating exploitability in a reasonable amount of time. We share our results in a table where each column indicates the quantile of performances across all these hyperparameter sets. For example, in the first table (Liar's Dice 1d6f) the value in the `rnad` row and `25%` column indicates that 25% of the runs achieved an exploitability of lower than 0.387.
> >
> > The trends appear similar to those of larger games.
> >
> > ### Liar's Dice 1d6f Exploitability Results
> > | Algorithm | 10% | 25% | 50% | 75% | 90% |
> > |:----------------------|:--------------|----------:|----------:|----------:|----------:|
> > | mmd | 0.090 | 0.128 | 0.174 | 0.296 | 0.412 |
> > | ppo | 0.086 | 0.142 | 0.168 | 0.332 | 0.412 |
> > | ppg | 0.074 | 0.129 | 0.183 | 0.355 | 0.678 |
> > | rnad | 0.373 | 0.387 | 0.441 | 0.516 | 0.660 |
> > | nfsp | 0.445 | 0.551 | 0.656 | 0.737 | 0.775 |
> > | escher | 0.417 | 0.524 | 0.560 | 0.626 | 0.844 |
> > | psro | 0.480 | 0.529 | 0.692 | 0.805 | 0.843 |
> >
> > ### Liar's Dice 1d5f Exploitability Results
> > | Algorithm | 10% | 25% | 50% | 75% | 90% |
> > |:----------------------|:--------------|----------:|----------:|----------:|----------:|
> > | mmd | 0.102 | 0.173 | 0.219 | 0.328 | 0.643 |
> > | ppo | 0.108 | 0.144 | 0.224 | 0.330 | 0.547 |
> > | ppg | 0.082 | 0.142 | 0.228 | 0.352 | 0.574 |
> > | rnad | 0.258 | 0.278 | 0.321 | 0.561 | 0.644 |
> > | nfsp | 0.253 | 0.382 | 0.562 | 0.733 | 0.733 |
> > | escher | 0.448 | 0.462 | 0.548 | 0.581 | 0.686 |
> > | psro | 0.364 | 0.427 | 0.532 | 0.720 | 0.756 |
> >
> > ### Liar's Dice 1d4f Exploitability Results
> > | Algorithm | 10% | 25% | 50% | 75% | 90% |
> > |:----------------------|:--------------|----------:|----------:|----------:|----------:|
> > | mmd | 0.056 | 0.100 | 0.183 | 0.300 | 0.610 |
> > | ppo | 0.055 | 0.096 | 0.178 | 0.326 | 0.616 |
> > | ppg | 0.063 | 0.097 | 0.180 | 0.340 | 0.563 |
> > | rnad | 0.157 | 0.180 | 0.206 | 0.272 | 0.412 |
> > | nfsp | 0.299 | 0.405 | 0.479 | 0.623 | 0.671 |
> > | escher | 0.368 | 0.442 | 0.500 | 0.554 | 0.592 |
> > | psro | 0.259 | 0.360 | 0.435 | 0.609 | 0.705 |
> >
> >
> > > Do you know of specific counterexamples where PPO/PPG fail to converge
> >
> > The notion of convergence is a bit difficult to pin down for deep RL, so it's a bit hard to answer. In terms of performance, we have not experienced an example in which they totally fail.
> >
> > > Is the plot in Figure 2 "the hyperparameter tuning launch" or the "evaluation launch"?
> >
> > It includes both. The tuning launch is the left box and the evaluation launch is the right box. Note that this is stated in both the caption and in text.

---

> > > ### Author Response · Authors · 2025-11-19
> > > **Author response (3/3)**
> > >
> > > > Do I understand correctly that you run PSRO with at most 8000 steps per oracle training?
> > >
> > > It's at most 8000 _episodes_ (not steps). These oracles are initialized from the parameters of the previous oracle though (this is the default behavior as implemented by OpenSpiel), so in practice they have more training.
> > >
> > > In case you were curious, we ran an additional sweep (new sweep), with the number of episodes per oracle varying from 2,500 to 160,000.
> > >
> > > | game                  | run           |   expl10% |   expl25% |   expl50% |   expl75% |
> > > |:----------------------|:--------------|----------:|----------:|----------:|----------:|
> > > | abrupt_dark_hex       | new sweep |  0.929014 |  1        |  1        |         1 |
> > > | abrupt_dark_hex       | sweep in paper |  0.750014 |  0.894206 |  1        |         1 |
> > > | abrupt_phantom_ttt    | new sweep |  0.934627 |  1        |  1        |         1 |
> > > | abrupt_phantom_ttt    | sweep in paper |  0.795704 |  0.893034 |  0.988603 |         1 |
> > > | classical_dark_hex    | new sweep |  0.969553 |  1        |  1        |         1 |
> > > | classical_dark_hex    | sweep in paper |  0.72258  |  0.862278 |  1        |         1 |
> > > | classical_phantom_ttt | new sweep |  0.88345  |  1        |  1        |         1 |
> > > | classical_phantom_ttt | sweep in paper |  0.786203 |  0.886317 |  0.99254  |         1 |
> > >
> > > Increasing the number of episodes per oracle yielded worse performance, as shown in the table.
> > >
> > > > Can you provide a compute time comparison between your BR implementations and the OpenSpiel one?
> > >
> > > Yes.
> > >
> > > We measure the run time of OpenSpiel's Best Response computation and compare it to ours for the games in the submission. The runtime was limited to an hour on a AMD EPYC 7413 24-Core Processors with 256 GB of memory,
> > >
> > > | Game | OpenSpiel BR Compute Time | Our BR Compute Time |
> > > |--------|-------------|---|
> > > | Liar's Dice 1d4f      | 1 s  | 0.03s|
> > > | Liar's Dice 1d5f       | 1 s | 0.03s|
> > > | Liar's Dice 1d6f       | 5s  |0.04s|
> > > | Liar's Dice 2d3f       | 10s  |0.04s|
> > > | Liar's Dice 2d4f       | 527s  |0.45s|
> > > | Liar's Dice 2d5f       |  Out of memory before timeout |11.3s|
> > > | Classical Phantom TTT      |  Out of memory before timeout |27s|
> > > | Classical Dark Hex      |  Out of memory before timeout |27s|
> > > | Abrupt Phantom TTT      | Out of memory before timeout  |47s|
> > > | Abrupt Dark Hex      |  Out of memory before timeout |47s|
> > >
> > > Note that Liar's dice calculation are single threaded while we use multiple threads for Hex & TTT variants

---

> > > > ### Comment · Reviewer_WVeg · 2025-11-24
> > > >
> > > > Thank you for the detailed rebuttal.
> > > >
> > > > I still very much dislike the very vague box being titled "hypothesis". I believe the scientific method is a key pilar of progress and it should not be bended and redefined. A hypothesis must be a testable prediction that can possibly be proven or falsified. I can come up with many nonsensical "policy gradient methods" that will "share ethos" with MMD, because they will, e.g., include the same terms, but with different signs, and they will not work better than CFR regardless of tuning. CFR does not strictly define how the local regret is minimized, so you may certainly interpret some of its variants as policy gradient. The box does not even specify in what measure the PG methods should be "superior". Please, consider rephrasing or removing the box and references to the "hypothesis".
> > > >
> > > >
> > > > Thank you for the experiments on smaller games. Am I right that they are showing the same story as the larger experiments? It reinforces the story that PG seems to work well in games. However, it also shows that scaling up the BR to larger games is not that important in this particular case. I believe you should include the experiments in the paper as well. I do not see them there at the moment.
> > > >
> > > > I would also be really curious to see the performance of the agent in biased RPS and matching pennies variants to test your "hypothesis".
> > > >
> > > > I suggest you update the caption to Figure 2. I was looking at it for some time before and I was looking for separate plots that would show the tunning and evaluation runs.
> > > >
> > > > Thank you for the BR comparison results. I do not see them in the revision. Do you plan to include them in the appendix to support the benefit of the BR you publish?

---

> > > > > ### Author Response · Authors · 2025-11-25
> > > > > **Author response 2**
> > > > >
> > > > > Thanks for your continued engagement.
> > > > >
> > > > > > Usage of the term hypothesis
> > > > >
> > > > > Subsequent to the initial review, we genuinely deliberated about possible other terms, but it is not obvious there is a better one. The term "Conjecture" has connotations of something that can be proven, "Ansatz" has some implication of a derivation to come, "Thesis" possibly has stronger intimations of an argument or a position to be defended.
> > > > >
> > > > > Merriam–Webster definition (1b) of hypothesis is "a predicted or anticipated outcome". The predicted outcome here is that tuned policy gradient algorithms that are reasonably similar to MMD will generally outperform algorithms based on FP, DO, and CFR in 2p0s IIGs. As the reviewer fairly points out, there is some amount of ambiguity on the boundary. But there nevertheless manifestly exist both experimental outcomes that would provide evidence for and against the hypothesis. Those in the submission provided evidence for the hypothesis. But counterfactually, if the policy gradient methods had been outperformed in our experiments by the FP-, DO-, and CFR-based methods, we would've considered the experimental outcome strong evidence that the hypothesis was false.
> > > > >
> > > > > And even though there is some boundary ambiguity, there are clearly situations in which having a sentiment for the veracity of this hypothesis would have practical ramifications (eg, a practitioner deciding what algorithm to try first for their problem).
> > > > >
> > > > > We would argue that a similar "ambiguity" criticism could be made of the hypothesis "Adam generally outperforms SGD"; yet, it is widely perceived to be true on the basis of experimental evidence and that perception has had significant practical ramifications.
> > > > >
> > > > > > Thank you for the experiments on smaller games. Am I right that they are showing the same story as the larger experiments? It reinforces the story that PG seems to work well in games.
> > > > >
> > > > > The relative performances are similar, yes.
> > > > >
> > > > > > However, it also shows that scaling up the BR to larger games is not that important in this particular case
> > > > >
> > > > > We believe this may blur the line between correlation and causation. In the context of deep reinforcement learning algorithms, the important factor is performance in large games. While performance on games that can be solved easily with tabular algorithms may sometimes correlate with performance in large games, it is not generally the case that methods optimized for very small games perform well at scale. There is a reason single-agent deep reinforcement learning algorithms are not typically benchmarked on tasks like Mountain Car and CartPole.
> > > > >
> > > > > > I suggest you update the caption to Figure 2. I was looking at it for some time before and I was looking for separate plots that would show the tuning and evaluation runs.
> > > > >
> > > > > Updated.
> > > > >
> > > > > > Thank you for the BR comparison results. I do not see them in the revision. Do you plan to include them in the appendix to support the benefit of the BR you publish?
> > > > >
> > > > > Yes, good idea. We updated the appendix.
> > > > >
> > > > > ---
> > > > >
> > > > > ### Possibility of Score Increase
> > > > > Since the initial review, there have been two corrected understandings and three new sets of experiments performed in response to/at the request of the review:
> > > > > 1. The corrected understanding about the fact that other works in this literature have not benchmarked exact exploitability on large games.
> > > > > 2. The corrected understanding that OpenSpiel does not (as a matter of practice) support exploitability for games of the size considered by the submission.
> > > > > 3. Additional experiments performed at the request of the reviewer on smaller games.
> > > > > 4. Additional experiments performed in response to the reviewer's question about the number of PSRO steps per oracle.
> > > > > 5. Additional experiments performed at the request of the reviewer about best response times.
> > > > >
> > > > > In light of this, especially because the corrected understandings are of relatively substantial significance, we respectfully ask whether the reviewer might consider increasing their assessed score.

---

> > > > > > ### Comment · Reviewer_WVeg · 2025-11-26
> > > > > >
> > > > > > Thank you for your response. I will not consider increasing the score unless we settle on a satisfactory phrasing of what seems to be the main message of the paper that you push. Since you seem to insist on the current phrasing, which I consider technically incorrect, I am considering lowering the score, not increasing.
> > > > > >
> > > > > > CFR is a wide family of methods that focus on minimizing counterfactual regret. You can minimize the regret in endless ways, direct or indirect, and you can prove about some PG methods that they minimize regret, so you can likely show that they minimize counterfactual regret in sequential games. It is a false dichotomy.
> > > > > >
> > > > > > Methods "based on FP or DO" is a little more specific, but almost anything that uses an approximation of best response can be seen as a method from this category.
> > > > > >
> > > > > > I would also have a problem with a "hypothesis" that "Adam generally outperforms SGD", but a more fitting  analogy to your "hypothesis" would IMO be "Methods that share ethos with Adam are comparable or superior to methods based on value estimates and mathematical programming". It is IMO just too vague and wide to claim in this form. It could mislead less experienced researchers and discourage them from looking at many relevant research directions.
> > > > > >
> > > > > > What I would be happy with is something like: "Previously published model-free deep reinforcement learning methods based on CFR, DO, FP converge to equilibria of zero-sum imperfect-information games slower than appropriately tuned MMD, PPO and PPG." Naturally, not only in the box buy also in similar claims elsewhere in the paper.

---

> ### Author Response · Authors · 2025-12-03
> **Final comments**
>
> We thank the reviewer for their continued engagement. We appreciate that we have different opinions.
>
> > CFR is a wide family of methods that focus on minimizing counterfactual regret. You can minimize the regret in endless ways, direct or indirect, and you can prove about some PG methods that they minimize regret, so you can likely show that they minimize counterfactual regret in sequential games. It is a false dichotomy.
>
> Policy gradient algorithms that use action-value feedback do not minimize counterfactual regret. (That is why new methods were required for CFR-based approaches.) So we respectfully disagree that it is a false dichotomy.
>
> > Regarding "Methods that share ethos with Adam are comparable or superior to methods based on value estimates and mathematical programming"
>
> We respectfully disagree this is an appropriate analogy. The FP/DO/CFR classes delineated by the submission are clearly divided, were created to address the same specific class of problems, and are frequently compared against one another in literature.
>
> > Regarding "wideness" of hypothesis and the possibility of discouraging researchers
>
> We agree with the reviewer that the hypothesis formulated by the submission is "wide". Because of this, and the contribution of our exact exploitability library, we have provided clear opportunities to falsify our hypothesis. We hope that this encourages researchers in this area.

---

### Official Review · Reviewer_rrhL · 2025-10-31

**Soundness:** 2
**Presentation:** 2
**Contribution:** 2
**Rating:** 4
**Confidence:** 4

**Summary:**

This paper puts forward a hypothesis that policy gradient (PG) methods based on (divergence-regularized) self-play can outperform reinforcement learning (RL) implementations of fictitious play (FP), double oracle (DO), or counterfactual regret minimization (CFR) in solving two-player zero-sum imperfect-information games. The authors empirically verify this hypothesis through extensive experimental analysis under benchmark games. Under exact exploitability computations or head-to-head evaluations, PG methods like magnetic mirror descent (MMD) consistently outperform FP/DO/CFR-based methods.

**Strengths:**

1. This paper makes significant efforts in aligning the existing game RL algorithms and fairly evaluating them under the strict exploitability measure.

2. The elaborations on the experimental settings and choice of benchmarks make the conclusion convincing.

**Weaknesses:**

1. My major concern about this paper is that the proposed hypothesis is not novel. To me, the hypothesis is somewhat equivalent to saying last-iterate convergence is superior to average-iterate convergence under current game RL implementions. I think it has been common sense since recent game-theoretic research has a significant focus on last-iterate convergence (e.g., [1,2]), which usually implies a linear convergence rate. In some small games, the last iterates can converge exponentially faster than the average iterates, even under the same learning dynamics (e.g., optimistic FTRL). The superiority of MMD in larger games is therefore predictable.

2. The benchmarks do not include real-world games like Battleship and No-Limit Texas Hold’em. To me, these games are more interesting because they could have more complicated transitivity structures in the policy space. Performance in these real-world games could be different from in the benchmark games considered in this paper. For example, it remains unclear whether MMD can also solve Stratego as R-NaD.

[1] Kenshi Abe, et al. Adaptively perturbed mirror descent for learning in games. ICML, 2024.

[2] Runyu Lu, et al. Divergence-regularized discounted aggregation: Equilibrium finding in multiplayer partially observable stochastic games. ICLR, 2025.

**Questions:**

Could you make a more detailed comparison between the implementations of MMD (based on mirror descent) and R-NaD (based on FTRL) to explain their performance discrepancy? Since gradient descent, multiplicative weights, Hedge, and FTRL have inherent relationships, MMD and R-NaD could be similarly viewed as a kind of regularized FTRL from the game-theoretic perspective.

---

> ### Author Response · Authors · 2025-11-19
> **Author response**
>
> > My major concern about this paper is that the proposed hypothesis is not novel. To me, the hypothesis is somewhat equivalent to saying last-iterate convergence is superior to average-iterate convergence under current game RL implementions. I think it has been common sense since recent game-theoretic research has a significant focus on last-iterate convergence (e.g., [1,2]), which usually implies a linear convergence rate. In some small games, the last iterates can converge exponentially faster than the average iterates, even under the same learning dynamics (e.g., optimistic FTRL). The superiority of MMD in larger games is therefore predictable.
>
> The hypothesis is not equivalent to saying last-iterate convergence is superior to average-iterate convergence. For example, as the reviewer points out, an important class of last-iterate algorithms is that engendered by optimism; the hypothesis does not pertain to such algorithms. Furthermore, it is important to note that existing theory does not predict which deep reinforcement learning algorithms will have superior performance in practice.
>
> > The benchmarks do not include real-world games like Battleship and No-Limit Texas Hold’em. To me, these games are more interesting because they could have more complicated transitivity structures in the policy space. Performance in these real-world games could be different from in the benchmark games considered in this paper.
>
> As discussed in the text, important criteria for benchmark games for this literature is that they support exact exploitability computations. There is no existing technique capable of doing this for no-limit Texas hold'em (due to its size). We would reiterate that the submission represents the largest games across which the exact exploitability of so many model-free algorithms have been compared (and even games of this size required us to build highly optimized implementations).
>
> > For example, it remains unclear whether MMD can also solve Stratego as R-NaD.
>
> This claim is incorrect both with respect to whether R-NaD solved Stratego (it did not) and with respect to whether it is unclear that MMD would perform well in Stratego (an approach based on it achieved the first superhuman result).
>
> https://arxiv.org/abs/2511.07312
>
> > Could you make a more detailed comparison between the implementations of MMD (based on mirror descent) and R-NaD (based on FTRL) to explain their performance discrepancy? Since gradient descent, multiplicative weights, Hedge, and FTRL have inherent relationships, MMD and R-NaD could be similarly viewed as a kind of regularized FTRL from the game-theoretic perspective.
>
> Yes. The standard implementation of R-NaD differs from the standard implementation of MMD in that R-NaD updates its reference policy, whereas MMD does not. The two also differ in that R-NaD uses neural replicator dynamics-style clipping, whereas MMD uses standard policy gradient techniques. In summary, though their tabular forms have inherent relationships, the means by which they are translated into deep reinforcement learning algorithms have important differences.

---

> > ### Comment · Reviewer_rrhL · 2025-11-19
> >
> > Thanks for the authors' rebuttal. I am generally satisfied with the response, especially with respect to the MMD and R-NaD comparison. Therefore, I am increasing the soundness from 2 to 3 and the overall rating from 4 to 6.

---

### Official Review · Reviewer_op4U · 2025-11-01

**Soundness:** 4
**Presentation:** 3
**Contribution:** 3
**Rating:** 8
**Confidence:** 4

**Summary:**

The paper revisits a common claim that policy gradient methods in self-play perform poorly in IIGs.  The paper does a thorough comparison of a set of 6 DRL methods for IIG (3 representative of adaptations of tabular methods with theoretical guarantees in tabular settings, and 3 policy gradient methods).

**Strengths:**

This is important work!  It's very important to be sure the community is not trapped by oft-repeated claims that are just known, but not truly validated.  The goals of this paper are important.  But, what is most impressive about the paper is the carefulness in the transparency of the comparison, recognition of their own biases, aim for reproducibility, and the limitations of what the empirical results justify.  This is commendable.

The following line is one of my favourites a paper focused on empirical results: "However, since we have hypothesized these algorithms will underperform, our role in tuning them should be carefully scrutinized. To aid readers in this endeavor, we provide detailed documentation of our algorithm implementations, hyperparameter tuning procedure, and hyperparameter tuning results, in addition to final results."

The section on limitations at the end is also valuable.  More papers should be quick to say what they aren't saying the way Section 6 does in the third sentence.

**Weaknesses:**

I think the biggest weakness of the paper is the paper comes at this work with a very strong bias and preconceived expectation to the result.  I am glad the authors recognize this bias, but I think that bias gives an edge ot the writing that is not needed in the text.  I believe the authors gave all algorithms a fair shake.  More importantly they are transparent about the shake they gave them.  But you don't get that picture through the first half of the paper.  I found myself immediately bristling at the line, "(and, not least, the disinclination of researchers toward showing baselines outperforming their algorithmic contributions)", thinking that the authors were just going to be doing the same thing but the other direction.  In the end I was pleasantly surprised by the care taken.  But, I would highly recommend the authors think about reframing the paper to take a more neutral objective tone (while still admitting their own hypothesis).  I would see value in this paper even if the results had gone the other way, and confirmed oft-repeated notions that have not been rigorously tested.

My only quibble with the experimental design was the forced choice of Adam to be used for all algorithms.  Do you have a clear table as to which of these 6 algorithms use Adam as the default optimizer?  That might be helpful to rule out that as a source of difference.

I'll use the rest of this section to make some suggestions about how the paper could possibly be further improved.

There's an awful lot of repetition in the statements and ideas.  I think the paper probably could be a page shorter and have the same impact.  Which would then open up space for some valuable additions.

I think it would be worth discussing what soundness results exist in this space.  I find the biased edge in writing seems to be dismissing trying to extend tabular methods with soundness guarantees into the world of parameterized approximations, which seems like an odd choice.  Wouldn't we want some theory to explain what works in practice?  I think it would be valuable to discuss what soundness guarantees we do have for any of the explored methods, both in tabular settings and for the particular application to either games (for PG methods) or function approximation (for other methods).  This would add considerably to clarifying the current state of the field.  The paper argues that policy gradient methods get "dismissed as unsound or relegated to the role of sacrificial baseline", but this paper doesn't offer anything to suggest that they are sound, right?  Now I suspect that the extensions of FP, DO, and CFR don't have guarantees outside tabular, but this clarity would be helpful.

I would also like to see more discussion about why you think the poor performance of policy gradient methods in IIGs was so prominent given your empirical results.  You pose early in the paper that maybe it has to do with the hyperparameter space, where MMD used more entropy regularization than typically used.  Well, was that the reason?  Can you conclude anything about the reason from your experiments?  Even if it's just saying, we didn't find the entropy regularlization parameter to be a significant factor in policy gradient methods performance, would add something.

It would be nice to see some discussion of whether the methods actually achieve low absolute exploitability.  Are any of these small enough to run a tabular method on?  Or could you add one game that is?  It would be nice to know if PG methods are actually doing a good job of finding low exploitability strategies or if the non-PG methods are doing a particular poor job, despite their tabular counterparts being very effective.  Again, it seems there is a missed opportunity to clarify what gaps still exist in the field.  Similarly, could you show trends of exploitability as the network size increases across methods?  Are of these methods able to see scaling with compute and make use of additional capacity to reduce exploitability?

I think another valuable addition would be a brief discussion about DeepStack, Libratus, and/or Student of Games.  These are methods that absolutely scale to large games, with NLHE being many orders of magnitude bigger than the games posed in the benchmarks, while also having some theory (with expected limitations due to approximation quality of network architectures).  They are also not model-free and add decision-time search as a key component, so they are outside the scope of the paper's claims.  But it still seems useful to discuss in order to give a more comprehensive picture.

I think that's the summary of my suggestions: the authors do a very good job of giving evidence for their stated main claim, but there's an opportunity to be much more... to give a comprehensive picture of where we as a field are at in algorithms for learning in IIGs.  This is a good paper, but that would be a great one.

**Questions:**

No questions; see weaknesses for suggestions.

---

> ### Author Response · Authors · 2025-11-19
> **Author response (1/2)**
>
> We appreciate the recognition of the transparency and qualifications of the work. We're especially glad the line about scrutinizing our role in tuning resonated. We thought hard about how to design a study like this given our dual role as hypothesizers of the underperformance of some of the algorithms and evaluators of those algorithms. We're happy this came through, even if the framing earlier in the paper suggested otherwise.
>
> > Regarding: "(and, not least, the disinclination of researchers toward showing baselines outperforming their algorithmic contributions)"
>
> Thanks for this feedback. We do feel it is important to consider that such a bias exists (similiarly to how we feel it is important to acknowledge that our work is subject to analogous biases) but are happy to rephrase it; we changed the line to the following (and also emphasize that the submission is not making any conclusions about this, only discussing possible explanations.).
>
> "and the structural disincentives to reporting strong results for baselines"
>
> Please let us know if the reviewer finds the above language suitable and if there is any other language the reviewer thinks would be worth changing.
>
> > Do you have a clear table as to which of these 6 algorithms use Adam as the default optimizer?
>
> The default optimizers of the source implementations are shown below.
>
> |  Algorithm       |Default Optimizer         |
> |------------------|---------------|
> | Escher    | Adam         |
> | MMD       | Adam            |
> | PPO        | Adam          |
> | PPG           | Adam           |
> | RNaD |   Adam          |
> | PSRO | SGD  |
> | NFSP | SGD  |
>
> PSRO and NFSP do use Adam in these implementations, but several follow works on the original papers have employed Adam (shown below).
>
> | Algorithm       | Optimizer | Paper links         |
> |---|--:|---:|
> |NFSP|Adam|[DeepFP](https://arxiv.org/pdf/1903.09569), [MC-NFSP](https://nitinkamra1992.github.io/pdf/deepfp.pdf)|
> |PSRO| Adam | [PPD-PSRO](https://arxiv.org/pdf/2306.16884), [XDO](https://arxiv.org/pdf/2103.06426)|
>
> > I find the biased edge in writing seems to be dismissing trying to extend tabular methods with soundness guarantees into the world of parameterized approximations, which seems like an odd choice. Wouldn't we want some theory to explain what works in practice? I think it would be valuable to discuss what soundness guarantees we do have for any of the explored methods, both in tabular settings and for the particular application to either games (for PG methods) or function approximation (for other methods). This would add considerably to clarifying the current state of the field. The paper argues that policy gradient methods get "dismissed as unsound or relegated to the role of sacrificial baseline", but this paper doesn't offer anything to suggest that they are sound, right? Now I suspect that the extensions of FP, DO, and CFR don't have guarantees outside tabular, but this clarity would be helpful.
>
> To clarify, we do not dismiss the idea of extending tabular methods with soundness guarantees to parameterized approximations. We think it is a reasonable line of research. Furthermore, we agree on the value of theory that explains what works in practice. But it is important to note that _there is no existing predictive theory of the performance of deep reinforcement learning algorithms_. The theory salient to this literature serves as inspiration rather than as a guaranteed foundation. Thus, we feel that it is important (and of complementary value) to provide check/balances on the "taking inspiration from theory" line of research—especially since the field has largely coalesced around it. One could characterize our "just seeing what works under rigorous evaluation" approach as an example of such a check/balance.
>
> > I would also like to see more discussion about why you think the poor performance of policy gradient methods in IIGs was so prominent given your empirical results. You pose early in the paper that maybe it has to do with the hyperparameter space, where MMD used more entropy regularization than typically used. Well, was that the reason? Can you conclude anything about the reason from your experiments? Even if it's just saying, we didn't find the entropy regularlization parameter to be a significant factor in policy gradient methods performance, would add something.
>
> Great question! We included a study of this in Section E (see Figure 7, page 20). The figure shows that, for our experiments, the entropy regularization parameter was _very_ significant in the performance of the policy gradient methods. Just changing the entropy regularization coefficient parameter from "normally sized" to large improved performance from poor to good. This doesn't by any means show that "the reason" for poor performance in earlier work was due to the entropy regularization parameter, but it certainly supports the idea that it could have been a factor.

---

> > ### Author Response · Authors · 2025-11-19
> > **Author response (2/2)**
> >
> > > It would be nice to see some discussion of whether the methods actually achieve low absolute exploitability. Are any of these small enough to run a tabular method on? Or could you add one game that is? It would be nice to know if PG methods are actually doing a good job of finding low exploitability strategies or if the non-PG methods are doing a particular poor job, despite their tabular counterparts being very effective. Again, it seems there is a missed opportunity to clarify what gaps still exist in the field.
> >
> > Yes, we implemented exact solvers for the games (and they are included in the package). See Figure 4 (page 16) and Table 2 (page 17). None of the deep reinforcement learning methods are competitive with tabular solvers in terms of exploitability, but this shouldn't be surprising, as achieving very low exploitabilities requires a level of precision to which function approximation isn't well suited.
> >
> > > Similarly, could you show trends of exploitability as the network size increases across methods? Are of these methods able to see scaling with compute and make use of additional capacity to reduce exploitability?
> >
> > The experiments we ran were very expensive (for our resources) in terms of CPU hours, so it would unfortunately be difficult for us to study performance with network size scaling.
> >
> > > I think another valuable addition would be a brief discussion about DeepStack, Libratus, and/or Student of Games. These are methods that absolutely scale to large games, with NLHE being many orders of magnitude bigger than the games posed in the benchmarks, while also having some theory (with expected limitations due to approximation quality of network architectures). They are also not model-free and add decision-time search as a key component, so they are outside the scope of the paper's claims. But it still seems useful to discuss in order to give a more comprehensive picture.
> >
> > Indeed, they do scale to large games; however, there is an important caveat. They only scale to large games when the amount of hidden information remains small (due to their use of public belief state and public belief state-adjacent objects). It would actually be very challenging (perhaps intractable) to run these algorithms in games like classical 3x3 dark hex (and certainly intractable in 4x4) because there is no public information. This shortcoming is part of what has motivated the development of model-free approaches which do not rely on public information.

---

> > > ### Comment · Reviewer_op4U · 2025-11-26
> > > **Disappointed**
> > >
> > > I am remain positive about the paper, but am disappointed by the author response.
> > >
> > > I think the weakness in the paper's prejudicial tone (a lack a better word for it) is significant.  I believe the other reviewers' negative reaction to the paper were largely driven by this tone.  I believe the impact of the paper on the community, which should be high, won't be because of this tone.
> > >
> > > The author response seems to largely dismiss it.  Any suggestion to provide a broader perspective is met with stubbornness.
> > >
> > > There is just no compelling reason not to discuss the theoretical guarantees associated with the methods being examined. There is no reason not to include a brief discussion of methods that incorporate search and exploiting public information.  Such a discussion ideally would point out their strengths (scaling, theory) and limitations (limited applicability that depends on the size of the belief space).  It's not clear at all that this limitation is fundamental even.  Why not include a discussion of the exact exploitability results in the main body of the paper?  And I'm not convinced that a network scaling study is infeasible.  You don't have to scale up your networks for such a study.  Scale them down and do they do worse on exploitability is just as interesting (and how do -FP, -DO, and CFR algorithms fare in comparison)?
> > >
> > > The one suggestion they do consider is rewriting one parenthetical comment.  For which, I was never recommending it be changed.  I was just noting how much it felt like they were being as blinded by prejudgement in the tone the paper takes as much as the works they were critiquing.  I don't have a strong opinion between their original version and the suggestion they ask about.
> > >
> > > Am I changing my mind about how strongly I support the paper?  I don't think so.  But, it's a missed opportunity to make an impactful paper that can direct future research more clearly.  I hope the authors reread their one initial positive review, and reconsider revising the paper to take a more objective stance in their presentation and add supporting material and discussion that would present a holistic view of field.

---

> ### Author Response · Authors · 2025-12-03
> **Final comments**
>
> First, we would like to apologize for any potential miscommunication. We certainly did not intend to be dismissive or come across as stubborn.
>
> > I think the weakness in the paper's prejudicial tone (a lack a better word for it) is significant ... The one suggestion they do consider is rewriting one parenthetical comment. For which, I was never recommending it be changed. I was just noting how much it felt like they were being as blinded by prejudgement in the tone the paper takes as much as the works they were critiquing. I don't have a strong opinion between their original version and the suggestion they ask about.
>
> We apologize about any miscommunication here. We understood the original review to be using the parenthetical as an example of language that the reviewer disliked. While we disagree that the paper is prejudicial in tone (as the reviewer wrote, perhaps there is a better word for what the reviewer means), we offered to change any of the language that the reviewer found objectionable (and make the same offer to the AC) and did change the language of the one instance the reviewer pointed out.
>
> > There is just no compelling reason not to discuss the theoretical guarantees associated with the methods being examined.
>
> We respectfully point out that the submission _does_ discuss the theoretical guarantees of fictitious play, double oracle, and counterfactual regret minimization (see lines 143, 154, & 169). For the deep reinforcement learning variants of these algorithms, there are no theoretical guarantees.
>
> > There is no reason not to include a brief discussion of methods that incorporate search and exploiting public information. Such a discussion ideally would point out their strengths (scaling, theory) and limitations (limited applicability that depends on the size of the belief space).
>
> We respectfully consider it a bit out of scope for a paper making a specific contribution pertaining to model-free methods to include a strengths and weaknesses discussion about public belief state methods. This consideration is consistent with other papers on model-free methods for IIGs, which do not include such discussions.
>
> > Why not include a discussion of the exact exploitability results in the main body of the paper
>
> We respectfully point out that the primary experimental results of the main body (Figure 2) are for exact exploitability. The provision and use of exact exploitability is mentioned on lines 70, 74, 80, 327, 332, 400, 402, 408, 410, & 467 of the main body.

---

### Meta-Review · Area_Chair_LrGS · 2025-12-23

**Summary:**

**Paper Summary**: This paper revisits the question of whether policy-gradient-based approaches can match the state-of-the-art performance of the imperfect-information games. The authors came to this question based on the recent results that showed the magnetic mirror-descent (MMD)- based approaches match the performance of counterfactual regret (CFR)-based algorithms for the tabular setting. The rationale is that since the PG approach resembles the MMD, it should match the performance of the state-of-the-art approaches as well. The paper tried out PPO-like approaches with the state-of-the-art approaches for DRL imperfect-information games, and showed that the performance is comparable for five such games.

**Reviewers' Concerns**: Reviewers appreciate the methodology and the outcome of the paper. Reviewers raised concerns about the game setups chosen, understanding of the performance of the PG and the MMD methods, and the difference with the existing approaches. The authors have addressed those in their rebuttals. Reviewer WVeg raised an important concern regarding the phrase 'hypothesis', which this paper has used to pose its research question. What the AC understood from the reviewer's point is that the paper should not overclaim its contributions. I partially agree with this observation. However, I also believe that the paper has clearly stated its limitations. Hence, I would not count it as a negative.

**AC's take**: The AC has gone over the paper and the discussions. The AC appreciates the contributions of the paper. The AC suggests that the authors should address the points raised by the reviewers. However, the AC also felt that the paper is on the borderline. The AC has a few suggestions to improve the paper.

The paper also did not compare with some relevant recent works [A1,A2], which are also trying to prove the same objective.  Further, the AC also suggests comparing with stronger CFR/PSRO methods. Also, it might be worthwhile to do the hyper-parameter tuning for the other approaches so that one can compare the best vs best across the algorithms.

[A1]. Kalogiannis, F. and Farina, G., Policy Gradient Methods Converge Globally in Imperfect-Information Extensive-Form Games. In OPT 2025: Optimization for Machine Learning.

[A2]. Liu, M., Farina, G. and Ozdaglar, A., 2024. A policy-gradient approach to solving imperfect-information games with iterate convergence. arXiv preprint arXiv:2408.00751.

**Reviewer Concerns:**

Reviewers raised concerns about the game setups chosen, understanding of the performance of the PG and the MMD methods, and the difference with the existing approaches. The authors have addressed those in their rebuttals. Reviewer WVeg raised an important concern regarding the phrase 'hypothesis', which this paper has used to pose its research question. What the AC understood from the reviewer's point is that the paper should not overclaim its contributions. I partially agree with this observation. However, I also believe that the paper has clearly stated its limitations. Hence, I would not count it as a negative.

**Reviewer Scores:**

One reviewer scored 8, and the two other scored 4. One reviewer increased their score to 6. The reviewer with the score 4 has issues with the phrase `hypothesis' and the claims made by the paper. However, I felt that the paper had clearly stated the scope of this work and the limitations of their claims.

---

### Decision · Program_Chairs · 2026-01-26

Accept (Poster)